# A Simple and Optimal Approach for Universal Online Learning with Gradient Variations

**Yu-Hu Yan, Peng Zhao, Zhi-Hua Zhou**
National Key Laboratory for Novel Software Technology, Nanjing University, China
School of Artificial Intelligence, Nanjing University, China
{yanyh, zhaop, zhouzh}@lamda.nju.edu.cn

## Abstract

We investigate the problem of universal online learning with gradient-variation regret. Universal online learning aims to achieve regret guarantees without the prior knowledge of the curvature of the online functions. Moreover, we study the problem-dependent gradient-variation regret as it plays a crucial role in bridging stochastic and adversarial optimization as well as game theory. In this work, we design a universal approach with the *optimal* gradient-variation regret simultaneously for strongly convex, exp-concave, and convex functions, thus addressing an open problem highlighted by Yan et al. [2023]. Our approach is *simple* since it is algorithmically efficient-to-implement with a two-layer online ensemble structure and only 1 gradient query per round, and theoretically easy-to-analyze with a novel and alternative analysis to the gradient-variation regret. Concretely, previous works on gradient variations require controlling the algorithmic stability, which is challenging and leads to sub-optimal regret and less efficient algorithm design. Our analysis overcomes this issue by using a Bregman divergence negative term from linearization and a useful smoothness property.

## 1 Introduction

Online convex optimization (OCO) models a sequential $T$-round game between an online learner and the environments [Hazan, 2016, Orabona, 2019]. In each round $t \in [T]$, the learner selects a decision $\mathbf{x}_t$ from a convex compact set $\mathcal{X} \subseteq \mathbb{R}^d$. Simultaneously, the environments adversarially choose a convex loss function $f_t : \mathcal{X} \mapsto \mathbb{R}$. Subsequently, the learner suffers a loss of $f_t(\mathbf{x}_t)$, receives feedback on the function $f_t(\cdot)$, and updates her decision to $\mathbf{x}_{t+1}$. In OCO, the learner aims to optimize the game-theoretical performance measure known as regret [Cesa-Bianchi and Lugosi, 2006], which is formally defined as

$$\text{REG}_T \triangleq \sum_{t=1}^{T} f_t(\mathbf{x}_t) - \min_{\mathbf{x} \in \mathcal{X}} \sum_{t=1}^{T} f_t(\mathbf{x}). \tag{1.1}$$

It represents the learner's excess cumulative loss compared with the best fixed comparator in hindsight.

In OCO, *function curvatures* play an important role in the best attainable regret bounds. Traditional studies examine three types of curvatures: convexity, exp-concavity, and strong convexity. Specifically, for convex functions, online gradient descent (OGD) achieves $\mathcal{O}(\sqrt{T})$ regret [Zinkevich, 2003]; for $\alpha$-exp-concave functions, online Newton step assuming $\alpha$ is known obtains $\mathcal{O}(\frac{d}{\alpha} \log T)$ regret [Hazan et al., 2007]; and for $\lambda$-strongly convex functions, OGD with known $\lambda$ attains $\mathcal{O}(\frac{1}{\lambda} \log T)$ [Hazan et al., 2007]. These results are shown to be minimax optimal [Ordentlich and Cover, 1998, Abernethy et al., 2008]. Recent studies further strengthen them by introducing *two levels of adaptivity*.

---

*Correspondence: Peng Zhao <zhaop@lamda.nju.edu.cn>

38th Conference on Neural Information Processing Systems (NeurIPS 2024).

Table 1: Comparison with existing results. The second column shows the regret bounds for strongly convex, exp-concave, and convex functions, following the $\mathcal{O}(\cdot)$-notation. Note that we only list universal guarantees related to the gradient variation $V_T$ or the time horizon $T$. Each gradient-variation bound can directly apply a corresponding small-loss regret in analysis, which is formally stated in Theorem 2 and omitted here for clarity. We treat the $\log \log T$ factor as a constant and omit it. "# Gradient" is the number of gradient queries in each round, where "1" represents exactly one gradient query. And "# Base" stands for the number of base learners.

| Works | Regret Bounds | | | Efficiency | |
|---|---|---|---|---|---|
| | Strongly Convex | Exp-concave | Convex | # Gradient | # Base |
| van Erven and Koolen [2016] | $d \log T$ | $d \log T$ | $\sqrt{T}$ | 1 | $\log T$ |
| Wang et al. [2019] | $\log T$ | $d \log T$ | $\sqrt{T}$ | 1 | $\log T$ |
| Zhang et al. [2022a] | $\log V_T$ | $d \log V_T$ | $\sqrt{T}$ | $\log T$ | $\log T$ |
| Yan et al. [2023] | $\log V_T$ | $d \log V_T$ | $\sqrt{V_T \log V_T}$ | 1 | $(\log T)^2$ |
| **Ours** | $\log V_T$ | $d \log V_T$ | $\sqrt{V_T}$ | 1 | $\log T$ |

**High-level Adaptivity to Unknown Curvatures.** Traditional studies require function curvature information in advance to select suitable algorithms for provable bounds. However, the information could be hard to access in real-world applications. To this end, a line of research aims to design a single *universal* algorithm that does not require the curvature information while achieving the same regret guarantees as if knowing it, achieving the adaptivity to unknown curvatures. The pioneering work of MetaGrad [van Erven and Koolen, 2016] proposed carefully designed surrogate functions and achieved $\mathcal{O}(\frac{d}{\alpha} \log T)$ for $\alpha$-exp-concave functions and $\mathcal{O}(\sqrt{T})$ for convex functions. Subsequently, Wang et al. [2019] obtained the optimal $\mathcal{O}(\frac{1}{\lambda} \log T)$ regret for $\lambda$-strongly convex functions, while maintaining the optimal rates in the other cases. Another remarkable progress of Zhang et al. [2022a] proposed a flexible framework with simplified analyses and further enhanced the minimax results using smoothness. We provide a detailed introduction of this work in Section 2.2.

**Low-level Adaptivity to Gradient Variation.** Although the regret guarantees based on the time horizon $T$ are optimal in the minimax sense, in this work we are interested in achieving the *gradient-variation* regret [Chiang et al., 2012, Yang et al., 2014], which replaces the dependence of the time horizon $T$ by the gradient variation quantity defined in the following:

$$V_T \triangleq \sum_{t=2}^{T} \sup_{\mathbf{x} \in \mathcal{X}} \|\nabla f_t(\mathbf{x}) - \nabla f_{t-1}(\mathbf{x})\|^2. \tag{1.2}$$

Under smoothness assumptions, the minimax regret can be improved to $\mathcal{O}(\frac{1}{\lambda} \log V_T)$, $\mathcal{O}(\frac{d}{\alpha} \log V_T)$, and $\mathcal{O}(\sqrt{V_T})$ for $\lambda$-strongly convex, $\alpha$-exp-concave, and convex functions, respectively. In this work, continuing previous gradient-variation online learning results [Zhao et al., 2020, 2024, Zhang et al., 2022b, Chen et al., 2023], we focus on the gradient-variation regret for the following reasons: *(i)* gradient-variation bounds safeguard the minimax guarantees. Besides, as demonstrated by Zhao et al. [2024], the gradient-variation regret is more fundamental than another well-known problem-dependent quantity known as the small loss $F_T \triangleq \min_{\mathbf{x} \in \mathcal{X}} \sum_{t \leq T} f_t(\mathbf{x})$ [Srebro et al., 2010, Orabona et al., 2012] since gradient-variation regret can imply small-loss bounds directly in analysis; *(ii)* the gradient variation plays a crucial role in bridging adversarial and stochastic optimization [Sachs et al., 2022]; and *(iii)* the gradient-variation regret can be used to achieve fast rates in multi-player games [Syrgkanis et al., 2015, Zhang et al., 2022b]. More detailed explanations of the importance of achieving such adaptivity are provided at the end of this section.

Motivated by the aforementioned two levels of adaptivity, we focus on the problem of achieving *universal gradient-variation* regret, i.e., designing a single universal approach with gradient-variation regret across different curvature types without the prior knowledge of them. For this problem, Zhang et al. [2022a] achieved partial results of $\mathcal{O}(\frac{1}{\lambda} \log V_T)$, $\mathcal{O}(\frac{d}{\alpha} \log V_T)$, $\mathcal{O}(\sqrt{T})$ for $\lambda$-strongly convex, $\alpha$-exp-concave, and convex functions, respectively. Subsequently, Yan et al. [2023] proposed a carefully designed three-layer online ensemble approach to stabilize the algorithm and improved the convex result to $\mathcal{O}(\sqrt{V_T \log V_T})$, achieving the first universal gradient-variation guarantee. Although optimal for strongly convex and exp-concave functions, their results still exhibit a gap with the optimal $\mathcal{O}(\sqrt{V_T})$ regret in the convex case. Here "optimal" refers to matching the best known results with curvature information since problem-dependent lower bound cannot be easily obtained. The only lower bound we are aware of is $\Omega(\sqrt{V_T})$ for convex functions [Yang et al., 2014, Remark 5].

To handle the uncertainty, *online ensemble* is commonly employed and proven effective in enhancing the robustness [Zhou, 2012, Zhao, 2021], such as adaptive regret minimization [Hazan and Seshadri, 2007, Daniely et al., 2015, Zhang et al., 2019], dynamic regret minimization [Zhang et al., 2018, Zhao et al., 2020, 2024], and universal online learning [van Erven and Koolen, 2016, Zhang et al., 2022a, Yan et al., 2023]. Concretely, an online ensemble algorithm contains multiple *base learners* for exploring the environments and a *meta learner* for ensemble. In universal online learning, base learners make guesses on the curvature information, and the meta learner tracks the best base learner (i.e., with the most accurate guess) on the fly.

Due to the deployment of the online ensemble framework, *computational efficiency* has become a point of concern, with two essential factors. The first is the *number of base learners*, because each independently runs an online learning algorithm that involves time-consuming gradient computations and projections. The second factor is the *number of gradient queries*, especially when the gradient evaluation is costly, e.g., in nuclear norm optimization [Ji and Ye, 2009] and mini-batch optimization [Li et al., 2014]. In universal online learning, an "efficient" algorithm is expected to adopt only $\mathcal{O}(\log T)$ base learners, which is inherent to the online ensemble design, and only 1 gradient query per round, matching the gradient query complexity of standard OGD. In terms of this metric, Zhang et al. [2022a] employed $\mathcal{O}(\log T)$ base learners, but required $\mathcal{O}(\log T)$ gradient queries per round. Yan et al. [2023] used only 1 gradient query per round but required $\mathcal{O}((\log T)^2)$ base learners (caused by their three-layer algorithm design), resulting in reduced efficiency.

**Results.** Motivated by the above considerations of optimality and efficiency, in this work, we propose a simple universal approach that achieves the *optimal* $\mathcal{O}(\frac{1}{\lambda} \log V_T)$, $\mathcal{O}(\frac{d}{\alpha} \log V_T)$, and $\mathcal{O}(\sqrt{V_T})$ regret simultaneously for $\lambda$-strongly convex, $\alpha$-exp-concave, and convex functions, and is *efficient* with one gradient query per round and $\mathcal{O}(\log T)$ base learners, resolving a major open problem highlighted by Yan et al. [2023]. We summarize our theoretical results in Theorem 1, and compare our results with existing ones in Table 1. Furthermore, we validate the effectiveness of our approach by: *(i)* showing that our universal gradient-variation regret directly implies the optimal universal small-loss regret in analysis without any algorithm modifications; and *(ii)* applying them to the stochastically extended adversarial (SEA) model [Sachs et al., 2022], an intermediate framework between stochastic and adversarial optimization. We achieve the same state-of-the-art guarantees as Chen et al. [2024], but without curvature information. The details are provided in Section 4.

**Techniques.** Our technical contributions include two simple and novel analyses. First, the key to gradient-variation regret is to analyze its empirical version, formally, $\|\nabla f_t(\mathbf{x}_t) - \nabla f_{t-1}(\mathbf{x}_{t-1})\|_2^2$. The previous approach to addressing this term involves controlling the algorithmic stability $\|\mathbf{x}_t - \mathbf{x}_{t-1}\|_2^2$, which is highly challenging in universal online learning, leading to sub-optimal results and less efficient algorithm design [Yan et al., 2023]. In this work, we overcome this issue via a novel analysis by a useful smoothness property and a Bregman divergence negative term from linearization, where the latter is inspired by the recent advance in stochastic optimization [Joulani et al., 2020]. Second, we adopt the surrogate functions proposed by Yan et al. [2023] to reduce the gradient query complexity, and provide a novel analysis for the empirical gradient variation based on the surrogates. A technical comparison with previous works is provided in Section 4.

**Organization.** The rest of the paper is structured as follows. Section 2 introduces preliminaries and a general framework for universal online learning. Section 3 presents our efficient approach with the optimal universal gradient-variation regret. Section 4 presents the implication and application of our results. Finally Section 5 concludes the paper. All the proofs can be found in the appendices.

## 2 Preliminaries

In this section, we introduce some preliminary knowledge, including notations, assumptions, definitions, and a general framework of universal online learning.

### 2.1 Notations, Assumptions, and Definitions

For simplicity, we use $\|\cdot\|$ for $\|\cdot\|_2$ by default and use $a \lesssim b$ or $a = \mathcal{O}(b)$ if there exists a constant $C < \infty$ such that $a/b \leq C$. In the following, we introduce the assumptions used in this work.

**Assumption 1** (Boundedness). *For any* $\mathbf{x}, \mathbf{y} \in \mathcal{X} \subseteq \mathbb{R}^d$, *the domain diameter satisfies* $\|\mathbf{x} - \mathbf{y}\| \leq D$, *and for* $t \in [T]$, *the gradient norm of the online functions is bounded as* $\|\nabla f_t(\mathbf{x})\| \leq G$.

**Assumption 2** (Smoothness). For each $t \in [T]$, the online function $f_t(\cdot)$ is $L$-smooth, i.e., $\|\nabla f_t(\mathbf{x}) - \nabla f_t(\mathbf{y})\| \leq L\|\mathbf{x} - \mathbf{y}\|$ holds for any $\mathbf{x}, \mathbf{y} \in \mathbb{R}^d$.

Both assumptions are common in the literature. Specifically, the boundedness assumption is widely used in OCO [Hazan, 2016]. And the smoothness assumption is essential for first-order algorithms in achieving the gradient-variation regret [Chiang et al., 2012]. Note that here Assumption 2 requires smoothness on the whole $\mathbb{R}^d$ space and can be relaxed to a slightly larger domain than $\mathcal{X}$, formally, $\mathcal{X}_+ \triangleq \{\mathbf{x} + \mathbf{b} \mid \mathbf{x} \in \mathcal{X}, \mathbf{b} \in {}^{G}/_{L} \cdot \mathbb{B}\}$, where $\mathbb{B} \triangleq \{\mathbf{x} \mid \|\mathbf{x}\| \leq 1\}$ is a unit ball. We defer the details of this relaxed smoothness requirement and its derivation to Appendix A. In the following, we provide formal definitions of strong convexity and exp-concavity.

**Definition 1.** *For any* $\mathbf{x}, \mathbf{y} \in \mathcal{X}$, *a function* $f(\cdot)$ *is* $\lambda$-*strongly convex if* $f(\mathbf{x}) - f(\mathbf{y}) \leq \langle \nabla f(\mathbf{x}), \mathbf{x} - \mathbf{y} \rangle - \frac{\lambda}{2} \cdot \|\mathbf{x} - \mathbf{y}\|^2$; $f(\cdot)$ *is* $\alpha$-*exp-concave if* $f(\mathbf{x}) - f(\mathbf{y}) \leq \langle \nabla f(\mathbf{x}), \mathbf{x} - \mathbf{y} \rangle - \frac{\alpha}{2} \cdot \langle \nabla f(\mathbf{x}), \mathbf{x} - \mathbf{y} \rangle^2$.

Note that the formal definition of $\beta$-exp-concavity is that $\exp(-\beta f(\cdot))$ is concave. Under Assumption 1, $\beta$-exp-concavity implies Definition 1 with $\alpha = \frac{1}{2} \cdot \min\{1/(4GD), \beta\}$ [Hazan, 2016, Lemma 4.3]. Therefore, we adopt Definition 1 as an alternative definition of exp-concavity for clarity.

## 2.2 A General Framework for Universal Online Learning

In this part, we present a general framework of universal online learning [Zhang et al., 2022a, Yan et al., 2023]. Formally, we study the problem where the learner lacks the prior knowledge of curvature information, including *(i)* curvature type: convexity, exp-concavity, or strong convexity; and *(ii)* curvature coefficient: exp-concavity $\alpha$ or strong convexity $\lambda$. Without loss of generality, we focus on the case of $\alpha, \lambda \in [1/T, 1]$. If $\alpha, \lambda < 1/T$, even the optimal minimax results — $\mathcal{O}(\frac{d}{\alpha} \log T)$ for exp-concave functions and $\mathcal{O}(\frac{1}{\lambda} \log T)$ for strongly convex functions [Hazan et al., 2007] – become linear in $T$, rendering the regret bounds vacuous. On the other hand, if $\alpha, \lambda > 1$, we can simply treat them as $\alpha, \lambda = 1$, only making the regret worsen by an ignorable constant factor. This simplification is also adopted by Zhang et al. [2022a], Yan et al. [2023].

To handle the uncertainty of curvatures, an online ensemble structure is usually employed, with multiple base learners exploring the environments and a meta learner tracking the best base learner. More specifically, to deal with the unknown curvature coefficients $\alpha$ and $\lambda$, we discretize them into the following candidate pool [Zhang et al., 2022a]:

$$\mathcal{H} \triangleq \{1/T, 2/T, 4/T, \ldots, 2^{n-1}/T\}, \tag{2.1}$$

whose size is $n = \lceil \log_2 T \rceil + 1 = \mathcal{O}(\log T)$. It can be proved that the discretized candidate pool $\mathcal{H}$ can approximate the continuous value of $\alpha$ or $\lambda$ with negligible constant errors. By doing this, it is natural to design three distinct groups of base learners:

 *(i)* *strongly convex* base learners: $n$ in total, each of which implements the algorithm for strongly convex functions with a guess $\lambda_i \in \mathcal{H}$ of the strong convexity coefficient $\lambda$;
 *(ii)* *exp-concave* base learners: $n$ in total, each of which implements the algorithm for exp-concave functions with a guess $\alpha_i \in \mathcal{H}$ of the exp-concavity coefficient $\alpha$;
 *(iii)* *convex* base learner: only one, it runs the algorithm for convex functions.

In total, there are $N \triangleq (2n+1) = \mathcal{O}(\log T)$ base learners with a two-layer structure, which is for now necessary in this problem. The best base learner is the one with the right guess of the curvature type and the closest guess of the curvature coefficient. Taking $\lambda$-strongly convex functions as an example, the guessed coefficient of the best base learner (indexed by $i^*$) satisfies $\lambda_{i^*} \leq \lambda \leq 2\lambda_{i^*}$.

Denoting by $\mathbf{x}_{t,i}$ the decision generated by the $i$-th base learner at the $t$-th round, $p_{t,i}$ the weight of the meta learner on the $i$-th base learner, an online ensemble method outputs the final decision as $\mathbf{x}_t = \sum_{i \in [N]} p_{t,i} \mathbf{x}_{t,i}$. This forms a general framework for universal online learning and it remains to select suitable algorithms and loss functions for the meta and base learners. We will illuminate our concrete algorithm design in Section 3.1 and present a more detailed description of our meta/base learners configurations in Appendix B.

---

**Algorithm 1** A Simple Approach for Universal Online Learning with Gradient Variations

---

**Input:** Meta learner $\mathcal{A}$, base learners $\{\mathcal{B}_i\}_{i\in[N]}$
 1: **Initialize**: $p_{1,i} = 1/N$ and $\mathbf{x}_{1,i}$ to be an arbitrary decision inside $\mathcal{X}$ for all $i \in [N]$
 2: **for** $t = 1$ **to** $T$ **do**
 3:     Submit $\mathbf{x}_t = \sum_{i\in[N]} p_{t,i}\mathbf{x}_{t,i}$, suffer $f_t(\mathbf{x}_t)$, and observe the gradient $\nabla f_t(\mathbf{x}_t)$
        *# Meta Update:*
 4:     $\mathcal{A}$ updates to $\boldsymbol{p}_{t+1} \in \Delta_N$ using the rule (B.5) with learning rate (B.6)
        *# Base Update:*
 5:     Construct surrogates $h_{t,i}^{\exp}(\cdot)$ $h_{t,i}^{\mathrm{sc}}(\cdot)$, $h_{t,i}^{\mathrm{c}}(\cdot)$ defined in (3.1) using only $\nabla f_t(\mathbf{x}_t)$
 6:     $\mathcal{B}_i$ updates to $\mathbf{x}_{t+1,i}$ using surrogates functions (3.1) and update rules (B.1)-(B.3) for $i \in [N]$
 7: **end for**

---

# 3 Our Approach

In this section, we present our approach for universal online learning with gradient-variation regret. Section 3.1 presents the overall procedure of our proposed algorithm. Subsequently, we outline our two key technical components: Section 3.2 presents a novel analysis to handle the empirical gradient variation, and Section 3.3 introduces surrogate functions to improve efficiency and provides a corresponding analysis for the empirical gradient variation defined on surrogates. We finally provide the optimal universal gradient-variation regret guarantees in Section 3.4.

## 3.1 Overall Algorithm

In this part, we present our simple approach for universal online learning with gradient variations, summarized in Algorithm 1. Basically, it is a two-layer online ensemble. Base learners are implemented using the preliminary configurations given in Section 2.2 and on carefully designed surrogate functions. The meta learner runs OPTIMISTIC-ADAPT-ML-PROD [Wei et al., 2016] on linearized losses. We specify the algorithmic details below, and a more detailed procedure in Appendix B.

In Line 3, the learner makes a weighted combination of the base learners' decisions $\{\mathbf{x}_{t,i}\}_{i\in[N]}$ using the meta learner's weights $\boldsymbol{p}_t = (p_{t,1}, \ldots, p_{t,N})$, submits the final decision $\mathbf{x}_t$, suffers a loss $f_t(\mathbf{x}_t)$, and receives a single $\nabla f_t(\mathbf{x}_t)$ as the gradient feedback, using *only* 1 gradient query per round.

**Meta Algorithm.** In Line 4, the meta learner uses OPTIMISTIC-ADAPT-ML-PROD [Wei et al., 2016] to update the weights by the following rule:

$$p_{t+1,i} \propto \varepsilon_{t,i}\cdot\exp(\varepsilon_{t,i}m_{t+1,i})\cdot W_{t,i}, \quad W_{t,i} = \left(W_{t-1,i} \cdot \exp\left(\varepsilon_{t-1,i}r_{t,i} - \varepsilon_{t-1,i}^2(r_{t,i} - m_{t,i})\right)\right)^{\frac{\varepsilon_{t,i}}{\varepsilon_{t-1,i}}} .$$

Specifically, denoting by $\ell_{t,i} \triangleq \langle\nabla f_t(\mathbf{x}_t), \mathbf{x}_{t,i}\rangle$ the loss of the $i$-th dimension, the meta algorithm inputs: $r_{t,i} = \langle\boldsymbol{\ell}_t, \boldsymbol{p}_t\rangle - \ell_{t,i}$, the instantaneous regret; $\varepsilon_{t,i}$, a time-varying learning rate; and $m_{t,i}$, an estimation of the true loss of the $t$-th round (the choice of optimisms will be shown later). The meta algorithm then outputs the weights $\boldsymbol{p}_{t+1} = (p_{t+1,1}, \ldots, p_{t+1,N})$ of the next round.

With appropriate learning rates (B.6), OPTIMISTIC-ADAPT-ML-PROD achieves an optimistic second-order bound of $\sum_{t\le T} r_{t,i} \le \mathcal{O}(\sqrt{\log N \sum_t (r_{t,i} - m_{t,i})^2} + \log N)$, where the $\log N$ factor is negligible since the base learner number $N$ equals $\mathcal{O}(\log T)$ and we can treat $\mathcal{O}(\log\log T)$ as a constant [Luo and Schapire, 2015]. The formal guarantee of OPTIMISTIC-ADAPT-ML-PROD is deferred to Lemma 2 in the appendix. In our problem, the instantaneous regret $r_{t,i} = \langle\boldsymbol{\ell}_t, \boldsymbol{p}_t\rangle - \ell_{t,i} = \langle\nabla f_t(\mathbf{x}_t), \mathbf{x}_t - \mathbf{x}_{t,i}\rangle$. Thus we choose $m_{t,i} = \langle\nabla f_{t-1}(\mathbf{x}_{t-1}), \mathbf{x}_t - \mathbf{x}_{t,i}\rangle$ for the convex base learner and $m_{t,i} = 0$ otherwise (i.e., for exp-concave and strongly convex base learners).[2] By doing this, we can upper-bound $\sum_t\langle\nabla f_t(\mathbf{x}_t), \mathbf{x}_t - \mathbf{x}_{t,i}\rangle$ by $\mathcal{O}(\sqrt{\sum_t\langle\nabla f_t(\mathbf{x}_t) - \nabla f_{t-1}(\mathbf{x}_{t-1}), \mathbf{x}_t - \mathbf{x}_{t,i}\rangle^2})$ for the convex base learner and by $\mathcal{O}(\sqrt{\sum_t\langle\nabla f_t(\mathbf{x}_t), \mathbf{x}_t - \mathbf{x}_{t,i}\rangle^2})$ otherwise. Later in Section 3.3, we illuminate how such results could benefit the final regret guarantees.

---

[2] Although $\mathbf{x}_t$ is unknown when using $m_{t,i}$, we only need the scalar value of $\langle\nabla f_{t-1}(\mathbf{x}_{t-1}), \mathbf{x}_t\rangle$, which is bounded and can be efficiently solved via a one-dimensional fixed-point problem of $\langle\nabla f_{t-1}(\mathbf{x}_{t-1}), \mathbf{x}_t(z)\rangle = z$. $\mathbf{x}_t$ is a function of $z$ because $\mathbf{x}_t$ relies on $p_{t,i}$, $p_{t,i}$ relies on $m_{t,i}$ and $m_{t,i}$ relies on $z$. Interested readers can refer to Section 3.3 of Wei et al. [2016] for more details.

**Base Algorithm.** In Line 5, we adopt carefully designed surrogate functions for different types of base learners to reduce the gradient query complexity [Yan et al., 2023]. Specifically, strongly convex, exp-concave, and the convex base learners run on the surrogate functions below respectively:

$$h_{t,i}^{\text{sc}}(\mathbf{x}) \triangleq \langle \nabla f_t(\mathbf{x}_t), \mathbf{x} \rangle + \frac{\lambda_i}{4}\|\mathbf{x} - \mathbf{x}_t\|^2, \quad h_{t,i}^{\text{exp}}(\mathbf{x}) \triangleq \langle \nabla f_t(\mathbf{x}_t), \mathbf{x} \rangle + \frac{\alpha_i}{4}\langle \nabla f_t(\mathbf{x}_t), \mathbf{x} - \mathbf{x}_t \rangle^2, \quad (3.1)$$

and $h_{t,i}^{\text{c}}(\mathbf{x}) \triangleq \langle \nabla f_t(\mathbf{x}_t), \mathbf{x} \rangle$, where $\lambda_i, \alpha_i$ are selected from the candidate coefficient pool $\mathcal{H}$ in (2.1). We emphasize that the surrogate functions require only 1 gradient query $\nabla f_t(\mathbf{x}_t)$ per round. Finally, in Line 6, the $i$-th base learner $\mathcal{B}_i$ updates the decision to $\mathbf{x}_{t+1,i}$ using optimistic online mirror descent (OOMD) [Rakhlin and Sridharan, 2013], which is general and covers many algorithms of interest, such as OGD and online Newton step. For each curvature type (convexity, exp-concavity, or strong convexity), we adopt a correspondingly configured OOMD as the base learner. For detailed update rules of differently configured OOMD, we refer readers to (B.1)-(B.3) in Appendix B.

As for previous works, Zhang et al. [2022a] adopted ADAPT-ML-PROD [Gaillard et al., 2014] as the meta learner, which does not incorporate optimisms and thus is impossible to achieve gradient-variation regret for convex functions, and operates on the original loss function $f_t(\cdot)$ for base learners, which leads to a less efficient gradient query complexity of $\mathcal{O}(\log T)$ per round. Yan et al. [2023] used a two-layer meta algorithm MSMWC-MASTER [Chen et al., 2021] as the meta learner, resulting in a three-layer ensemble structure, which is also not efficient enough. Compared with approaches above, our Algorithm 1 is simpler and more efficient as it requires $\mathcal{O}(\log T)$ base learners and only 1 gradient query in each round. We emphasize that our contributions mainly lie in the technical aspects showing that although simple, our approach can achieve the optimal universal gradient-variation regret, which is accomplished via two novel analytical components.

### 3.2 Novel Analysis on Empirical Gradient Variations

In this part, we provide a novel analysis of the gradient-variation regret. For clarity, we illustrate from the lowest level — as we only use one gradient $\nabla f_t(\mathbf{x}_t)$ in the $t$-th round, to obtain the gradient variation $V_T$ defined in (1.2), it is necessary to first attain its *empirical* version $\bar{V}_T \triangleq \sum_{t \leq T} \|\nabla f_t(\mathbf{x}_t) - \nabla f_{t-1}(\mathbf{x}_{t-1})\|^2$. Previous studies decompose this term into two parts:

$$\|\nabla f_t(\mathbf{x}_t) - \nabla f_{t-1}(\mathbf{x}_{t-1})\|^2 \lesssim \|\nabla f_t(\mathbf{x}_t) - \nabla f_{t-1}(\mathbf{x}_t)\|^2 + \|\nabla f_{t-1}(\mathbf{x}_t) - \nabla f_{t-1}(\mathbf{x}_{t-1})\|^2$$
$$\leq \sup_{\mathbf{x} \in \mathcal{X}} \|\nabla f_t(\mathbf{x}) - \nabla f_{t-1}(\mathbf{x})\|^2 + L^2\|\mathbf{x}_t - \mathbf{x}_{t-1}\|^2,$$

using smoothness (i.e., Assumption 2). Aggregating the first term over $T$ rounds leads to the desired $V_T$ quantity and the remaining challenge is to control the algorithmic stability $\|\mathbf{x}_t - \mathbf{x}_{t-1}\|^2$. Consequently, since each decision is a weighted combination of base learners' decisions (i.e., $\mathbf{x}_t = \sum_{i \leq N} p_{t,i}\mathbf{x}_{t,i}$), the algorithmic stability is difficult to control. To this end, Yan et al. [2023] decomposed the stability term in the following way:

$$\|\mathbf{x}_t - \mathbf{x}_{t-1}\|^2 \lesssim \sum_{i=1}^{N} p_{t,i}\|\mathbf{x}_{t,i} - \mathbf{x}_{t-1,i}\|^2 + \|\boldsymbol{p}_t - \boldsymbol{p}_{t-1}\|_1^2. \quad (3.2)$$

Consequently, for the first term, the authors injected correction terms to the meta learner following Zhao et al. [2024]. To cancel the second term, the meta algorithm must include a corresponding negative stability term in its analysis, while achieving an optimistic second-order bound simultaneously. To the best of our knowledge, the only feasible algorithm satisfying both requirements is the two-layer meta algorithm MSMWC-MASTER [Chen et al., 2021], which leads to a three-layer online ensemble structure and therefore affects the efficiency. Besides, it attains a second-order bound of the form $\mathcal{O}(\sqrt{Q_{T,i} \log Q_{T,i}})$, where $Q_{T,i} \triangleq \sum_t (\ell_{t,i} - m_{t,i})^2$, which causes the sub-optimality of the regret guarantees with an additional logarithmic factor in the results of Yan et al. [2023].

In this work, we handle the empirical gradient variation alternatively via a novel and simple analysis with two key parts: *(i)* a negative term arising from linearization; and *(ii)* a useful smoothness property. First, we observe that the instantaneous regret can be transformed as:

$$f_t(\mathbf{x}_t) - f_t(\mathbf{x}^\star) = \langle \nabla f_t(\mathbf{x}_t), \mathbf{x}_t - \mathbf{x}^\star \rangle - \mathcal{D}_{f_t}(\mathbf{x}^\star, \mathbf{x}_t), \quad (3.3)$$

where $\mathbf{x}^\star \in \arg\min_{\mathbf{x} \in \mathcal{X}} \sum_t f_t(\mathbf{x})$ and $\mathcal{D}_f(\mathbf{x}, \mathbf{y}) \triangleq f(\mathbf{x}) - f(\mathbf{y}) - \langle \nabla f(\mathbf{y}), \mathbf{x} - \mathbf{y} \rangle$ is the Bregman divergence associated with function $f(\cdot)$. The last term is a *negative term from linearization*, which

can be seen as the compensation by treating a convex function as a linear one. Previous studies on the gradient-variation regret usually omit this term, while we show below that this negative term helps to achieve a much simpler analysis of the empirical gradient variation. Second, we introduce *a useful property of smoothness*, formally introduced below.

**Proposition 1** (Theorem 2.1.5 of Nesterov [2018]). *$f(\cdot)$ is $L$-smooth over $\mathbb{R}^d$ if and only if*

$$\|\nabla f(\mathbf{x}) - \nabla f(\mathbf{y})\|^2 \leq 2L \cdot \mathcal{D}_f(\mathbf{y}, \mathbf{x}), \quad \text{for any } \mathbf{x}, \mathbf{y} \in \mathbb{R}^d. \tag{3.4}$$

Compared with the commonly used $\|\nabla f(\mathbf{x}) - \nabla f(\mathbf{y})\| \leq L\|\mathbf{x} - \mathbf{y}\|$, Proposition 1 gives a tighter bound for the *squared* gradient changes since $\|\nabla f(\mathbf{x}) - \nabla f(\mathbf{y})\|^2 \leq 2L\mathcal{D}_f(\mathbf{y}, \mathbf{x}) \leq L^2\|\mathbf{x} - \mathbf{y}\|^2$, where the second step is due to $\mathcal{D}_f(\mathbf{y}, \mathbf{x}) \leq \frac{L}{2}\|\mathbf{x} - \mathbf{y}\|^2$, for any $\mathbf{x}, \mathbf{y} \in \mathbb{R}^d$ [Nesterov, 2018, Theorem 2.1.5], intuitively making the analysis easier. Combining the Bregman divergence negative term (3.3) and this useful property (3.4), we address the empirical gradient variation effectively as

$$\bar{V}_T \lesssim \sum_{t=2}^{T} \left( \|\nabla f_t(\mathbf{x}_t) - \nabla f_t(\mathbf{x}^\star)\|^2 + \|\nabla f_t(\mathbf{x}^\star) - \nabla f_{t-1}(\mathbf{x}^\star)\|^2 + \|\nabla f_{t-1}(\mathbf{x}^\star) - \nabla f_{t-1}(\mathbf{x}_{t-1})\|^2 \right)$$

$$\overset{(3.4)}{\lesssim} L\sum_{t=2}^{T}\mathcal{D}_{f_t}(\mathbf{x}^\star, \mathbf{x}_t) + V_T + L\sum_{t=2}^{T}\mathcal{D}_{f_{t-1}}(\mathbf{x}^\star, \mathbf{x}_{t-1}) \leq 2L\sum_{t=1}^{T}\mathcal{D}_{f_t}(\mathbf{x}^\star, \mathbf{x}_t) + V_T, \tag{3.5}$$

where the first step introduces intermediate terms $\nabla f_t(\mathbf{x}^\star)$ and $\nabla f_{t-1}(\mathbf{x}^\star)$, the second step uses Proposition 1, and the last step combines two summations into one by shifting the indexes of $t$. The Bregman divergence negative term in (3.3) can cancel the positive term in (3.5), leaving only the gradient variation quantity $V_T$ as desired.

Here we only require $\|\nabla f(\mathbf{x}) - \nabla f(\mathbf{y})\|^2 \leq 2L\mathcal{D}_f(\mathbf{y}, \mathbf{x})$ on $\mathcal{X}$ rather than $\mathbb{R}^d$, which can be satisfied by requiring $L$-smoothness only on a slightly larger domain than $\mathcal{X}$ (a relaxation of Assumption 2). Interested readers can refer to Appendix A for details. Finally we end this part with several remarks.

**Remark 1.** We emphasize that the Bregman divergence negative term comes from the linearization of convex functions, and is thus *algorithm-independent*. Therefore, we can eliminate the need to control the algorithmic stability, in contrast to previous works for gradient-variation regret [Chiang et al., 2012, Yan et al., 2023]. To the best of our knowledge, this is the *first* alternative analysis of the gradient-variation regret since first proposed by Chiang et al. [2012]. ◁

**Remark 2.** Our idea of the negative term from linearization is inspired by the recent advance in stochastic optimization [Joulani et al., 2020]. Note that they focus on a different problem of achieving the $\mathcal{O}(1/T^2)$ rate as Nesterov's accelerated gradient [Nesterov, 2018], while we investigate the gradient-variation regret in the universal online (adversarial) convex optimization setup. ◁

**Remark 3.** Our analysis does not strictly outperform those that directly handle the stability term as in (3.2), e.g., Yan et al. [2023], since the latter can be used for fast rates in the multi-player game setup [Syrgkanis et al., 2015, Zhang et al., 2022b]. A more detailed discussion of the advantages and disadvantages over previous approaches is provided in Section 4.3. ◁

### 3.3 Novel Analysis on Empirical Gradient Variations of Surrogates

Section 3.2 already suffices to achieve the optimal universal gradient-variation regret if *multiple* gradient queries are permitted. In this part, we consider further improving the computational efficiency by using only 1 gradient query per round, achieving the same gradient query complexity as OGD.

As stated in Section 3.1, we implement base algorithms on carefully designed surrogate functions following Yan et al. [2023]. In this part, we show that additional novel analysis is required to handle the empirical gradient variation *defined on surrogates*. To see this, we first provide the entire regret decomposition to give readers a clear roadmap. Specifically, taking $\lambda$-strong convexity as an example, the regret can be decomposed as follows:

$$\text{REG}_T \leq \sum_{t=1}^{T}\langle\nabla f_t(\mathbf{x}_t), \mathbf{x}_t - \mathbf{x}^\star\rangle - \frac{\lambda}{4}\sum_{t=1}^{T}\|\mathbf{x}_t - \mathbf{x}^\star\|^2 - \frac{1}{2}\sum_{t=1}^{T}\mathcal{D}_{f_t}(\mathbf{x}^\star, \mathbf{x}_t)$$

$$\leq \left[\sum_{t=1}^{T} r_{t,i^\star} - \frac{\lambda_{i^\star}}{4}\sum_{t=1}^{T}\|\mathbf{x}_t - \mathbf{x}_{t,i^\star}\|^2\right] + \left[\sum_{t=1}^{T}\left(h_{t,i^\star}^{\text{sc}}(\mathbf{x}_{t,i^\star}) - h_{t,i^\star}^{\text{sc}}(\mathbf{x}^\star)\right)\right] - \frac{1}{2}\sum_{t=1}^{T}\mathcal{D}_{f_t}(\mathbf{x}^\star, \mathbf{x}_t),$$

where the first step uses (3.3) and the fact that $\mathcal{D}_{f_t}(\mathbf{x}^\star, \mathbf{x}_t) \geq \frac{\lambda}{2}\|\mathbf{x}_t - \mathbf{x}^\star\|^2$ since $f_t(\cdot)$ is $\lambda$-strongly convex. The second step uses the definition of the best base learner (indexed by $i^\star$): $\lambda_{i^\star} \leq \lambda \leq 2\lambda_{i^\star}$ and the definition of surrogate functions defined in (3.1). The first term above (*meta regret*) assesses how well the algorithm tracks the best base learner, and the second term (*base regret*) measures the best base learner's performance. The meta regret contains a linearized regret with a negative term from curvatures. This negative term is useful for exp-concave and strongly convex functions if the linearized regret enjoys a second-order bound:

$$\sum_{t=1}^{T} r_{t,i^\star} - \frac{\lambda_{i^\star}}{4}\sum_{t=1}^{T}\|\mathbf{x}_t - \mathbf{x}_{t,i^\star}\|^2 \lesssim \sqrt{\sum_{t=1}^{T}\langle\nabla f_t(\mathbf{x}_t), \mathbf{x}_t - \mathbf{x}_{t,i^\star}\rangle^2} - \frac{\lambda_{i^\star}}{4}\sum_{t=1}^{T}\|\mathbf{x}_t - \mathbf{x}_{t,i^\star}\|^2 \lesssim \frac{1}{\lambda},$$

where the $1/\lambda$ factor can be absorbed by the final regret $\mathcal{O}(\frac{1}{\lambda}\log V_T)$. The base regret is defined on the surrogates, which preserves the curvature properties, but using only 1 gradient $\nabla f_t(\mathbf{x}_t)$.

Below we explain the reason for handling the empirical gradient variation defined on surrogates. Taking the $i$-th strongly convex base learner as an example, it updates as $\mathbf{x}_{t,i} = \Pi_{\mathcal{X}}[\widehat{\mathbf{x}}_{t,i} - \eta_t\mathbf{m}_t]$ and $\widehat{\mathbf{x}}_{t+1,i} = \Pi_{\mathcal{X}}[\widehat{\mathbf{x}}_{t,i} - \eta_t\nabla h_{t,i}^{\mathrm{sc}}(\mathbf{x}_{t,i})]$, an initialization of the OOMD algorithm, where $\eta_t$ represents the step size, $\Pi_{\mathcal{X}}[\mathbf{x}] = \arg\min_{\mathbf{y}\in\mathcal{X}}\|\mathbf{x} - \mathbf{y}\|$ denotes the Euclidean projection onto $\mathcal{X}$, and $\widehat{\mathbf{x}}_{t,i}$ is an intermediate variable. With appropriately chosen step sizes, the base learner achieves an optimistic bound of $\mathcal{O}(\log D_T)$, where $D_T = \sum_{t\leq T}\|\nabla h_{t,i}^{\mathrm{sc}}(\mathbf{x}_{t,i}) - \mathbf{m}_t\|^2$ (e.g., please refer to Theorem 15 of Chiang et al. [2012]). Therefore, choosing the optimism as $\mathbf{m}_t = \nabla h_{t-1,i}^{\mathrm{sc}}(\mathbf{x}_{t-1,i})$ leads to an empirical gradient-variation bound $\mathcal{O}(\log D_T)$ *defined on surrogates*,[3] where

$$D_T = \sum_{t=2}^{T}\|\nabla h_{t,i}^{\mathrm{sc}}(\mathbf{x}_{t,i}) - \nabla h_{t-1,i}^{\mathrm{sc}}(\mathbf{x}_{t-1,i})\|^2$$

$$= \sum_{t=2}^{T}\left\|\nabla f_t(\mathbf{x}_t) - \nabla f_{t-1}(\mathbf{x}_{t-1}) + \frac{\lambda_i}{2}(\mathbf{x}_{t,i} - \mathbf{x}_t) - \frac{\lambda_i}{2}(\mathbf{x}_{t-1,i} - \mathbf{x}_{t-1})\right\|^2.$$

The empirical gradient variation defined on the original functions, i.e., $\|\nabla f_t(\mathbf{x}_t) - \nabla f_{t-1}(\mathbf{x}_{t-1})\|^2$, can be handled via the analysis in Section 3.2. The main challenge is to deal with the rest terms caused by the surrogate functions. Yan et al. [2023] overcame this issue by controlling $(\mathbf{x}_t - \mathbf{x}_{t-1})$ and $(\mathbf{x}_{t,i} - \mathbf{x}_{t-1,i})$ separately. Again, as we have explained in Section 3.2, since the decision $\mathbf{x}_t$ is a weighted combination of base learners' decisions (i.e., $\mathbf{x}_t = \sum_{i\leq N} p_{t,i}\mathbf{x}_{t,i}$), handling the algorithmic stability term $\|\mathbf{x}_t - \mathbf{x}_{t-1}\|^2$ directly using (3.2) would results in a less efficient three-layer online ensemble structure and the sub-optimality of the regret guarantees, as Yan et al. [2023] did.

In this work, we propose a novel analysis — while controlling $(\mathbf{x}_{t,i} - \mathbf{x}_t) - (\mathbf{x}_{t-1,i} - \mathbf{x}_{t-1})$ in each individual round is hard, it can be bounded when *aggregated over the time horizon*. Specifically, we bound it by combining two summations into one:

$$\sum_{t=2}^{T}\|(\mathbf{x}_{t,i} - \mathbf{x}_t) - (\mathbf{x}_{t-1,i} - \mathbf{x}_{t-1})\|^2 \lesssim \sum_{t=2}^{T}\|\mathbf{x}_{t,i} - \mathbf{x}_t\|^2 + \sum_{t=2}^{T}\|\mathbf{x}_{t-1,i} - \mathbf{x}_{t-1}\|^2 \leq 2\sum_{t=1}^{T}\|\mathbf{x}_{t,i} - \mathbf{x}_t\|^2.$$

The same idea is also used in the derivation of (3.5). Consequently, this term can be canceled out by the negative term from curvatures in the meta regret. For this cancellation to occur, appropriate coefficients are chosen, which are provided in the detailed proofs (e.g., the 'Regret Analysis' part in the proof of Theorem 1) and are omitted here for clarity.

This simple and novel analysis eliminates the need to control the overall algorithmic stability term of $\|\mathbf{x}_t - \mathbf{x}_{t-1}\|^2$ required by previous works, and is essential for achieving the improved computational efficiency and the optimal regret guarantees, as shown in the next part.

### 3.4 Optimal Universal Gradient-Variation Regret Guarantees

In this part, we present our main theoretical result — our simple and efficient Algorithm 1 (in Section 3.1) which adopts two novel analyses (in Section 3.2 and Section 3.3) achieves the *optimal*

---

[3]For strongly convex functions, it is possible to choose $\mathbf{m}_t = \nabla f_{t-1}(\mathbf{x}_{t-1})$ to avoid additional surrogate-induced terms, that will be discussed below. We choose the gradient of the last round as the optimism since this is the the only choice at present to achieve an optimistic regret for exp-concave functions [Chiang et al., 2012].

gradient-variation regret *without* requiring the curvature information in advance for universal online learning. The corresponding proof is provided in Appendix C.

**Theorem 1.** *Under Assumptions 1 and 2 (or the relaxed Assumption 3), Algorithm 1 achieves $\mathcal{O}(\log V_T)$, $\mathcal{O}(d \log V_T)$, and $\mathcal{O}(\sqrt{V_T})$ for strongly convex, exp-concave, and convex functions.*

Theorem 1 improves the $\mathcal{O}(\sqrt{V_T \log V_T})$ bound of Yan et al. [2023] and is optimal by matching the best known results when the curvature information is known. It performs well when the gradient variation is small, such as $f_1 = f_2 = \cdots = f_T$ (where $V_T = 0$). Note that for $\alpha$-exp-concave or $\lambda$-strongly convex functions, our guarantee is actually $\mathcal{O}(\min\{\frac{d}{\alpha} \log V_T, \sqrt{V_T}\})$ or $\mathcal{O}(\min\{\frac{1}{\lambda} \log V_T, \sqrt{V_T}\})$, thus ensuring $\mathcal{O}(\sqrt{V_T})$ even when $\alpha = \mathcal{O}(1/T)$ or $\lambda = \mathcal{O}(1/T)$. This is because exp-concave and strongly convex functions are also convex and thus our convex bound is still applicable.

# 4 Implication, Application, and Discussion

In this section, we validate the effectiveness of our results by the implication of small-loss regret and the application in the SEA model. We also discuss the technical comparison with the previous correction-based approach [Yan et al., 2023] at the end of this section.

## 4.1 Implication to Universal Small-Loss Regret

In this part, we illustrate that our universal gradient-variation regret in Theorem 1 implies the universal small-loss regret measured by $F_T \triangleq \min_{\mathbf{x} \in \mathcal{X}} \sum_{t \leq T} f_t(\mathbf{x})$ *directly in analysis*, i.e., without any algorithmic modification, and thus safeguards the case of $F_T \leq V_T$, such as $\min_{\mathbf{x} \in \mathcal{X}} f_t(\mathbf{x}) = 0$ for any $t \in [T]$ (where $F_T = 0$). The corresponding proof is provided in Appendix D.1.

**Theorem 2.** *Under Assumptions 1 and 2 (or the relaxed Assumption 3), if the online functions are non-negative, Algorithm 1 achieves $\mathcal{O}(\log F_T)$, $\mathcal{O}(d \log F_T)$, and $\mathcal{O}(\sqrt{F_T})$ for strongly convex, exp-concave, and convex functions, respectively.*

Theorem 2 achieves the same optimal small-loss bounds as Zhang et al. [2022a]. Combined with Theorem 1, our approach achieves the *best known* problem-dependent regret guarantees in the universal online learning problem. In the end, we emphasize again that our approach is efficient as it requires $\mathcal{O}(\log T)$ base learners and only 1 gradient query in each round.

## 4.2 Application to Stochastically Extended Adversarial (SEA) Model

Stochastically extended adversarial (SEA) model [Sachs et al., 2022] interpolates between stochastic and adversarial online convex optimization. Formally, it assumes that the online function $f_t(\cdot)$ is sampled stochastically from an adversarially chosen distribution $\mathfrak{D}_t$. Denoting by $F_t(\cdot) \triangleq \mathbb{E}_{f_t \sim \mathfrak{D}_t}[f_t(\cdot)]$ the expected function, two terms capture the essential characteristics of SEA model:

$$\sigma_{1:T}^2 \triangleq \sum_{t=1}^{T} \max_{\mathbf{x} \in \mathcal{X}} \mathbb{E}_{f_t \sim \mathfrak{D}_t} \left[ \|\nabla f_t(\mathbf{x}) - \nabla F_t(\mathbf{x})\|^2 \right], \Sigma_{1:T}^2 \triangleq \mathbb{E} \left[ \sum_{t=2}^{T} \sup_{\mathbf{x} \in \mathcal{X}} \|\nabla F_t(\mathbf{x}) - \nabla F_{t-1}(\mathbf{x})\|^2 \right],$$

where $\sigma_{1:T}^2$ is the variance in sampling $f_t(\cdot)$ from $\mathfrak{D}_t(\cdot)$ and $\Sigma_{1:T}^2$ is the variation of $\{F_t(\cdot)\}_{t \in [T]}$. Sachs et al. [2022] initiated the study of the SEA model. For smooth expected functions $\{F_t(\cdot)\}_{t=1}^{T}$, they achieved the optimal $\mathcal{O}(\sqrt{\sigma_{1:T}^2 + \Sigma_{1:T}^2})$ regret for convex expected functions, and $\mathcal{O}((\sigma_{\max}^2 + \Sigma_{\max}^2) \log T)$ in the strongly convex case, where $\sigma_{\max}^2 \triangleq \max_{t \in [T]} \max_{\mathbf{x} \in \mathcal{X}} \mathbb{E}_{f_t \sim \mathfrak{D}_t}[\|\nabla f_t(\mathbf{x}) - \nabla F_t(\mathbf{x})\|^2]$ and $\Sigma_{\max}^2 \triangleq \max_{t \in [T]} \sup_{\mathbf{x} \in \mathcal{X}} \|\nabla F_t(\mathbf{x}) - \nabla F_{t-1}(\mathbf{x})\|^2$. Subsequently, Chen et al. [2024] improved the strongly convex regret to $\mathcal{O}((\sigma_{\max}^2 + \Sigma_{\max}^2) \log((\sigma_{1:T}^2 + \Sigma_{1:T}^2)/(\sigma_{\max}^2 + \Sigma_{\max}^2)))$ and obtained $\mathcal{O}(d \log(\sigma_{1:T}^2 + \Sigma_{1:T}^2))$ regret for exp-concave individual functions $\{f_t(\cdot)\}_{t=1}^{T}$.

The gradient variation is essential in connecting the stochastic and adversarial optimization [Chen et al., 2023, Lemma 4], which is also restated in (D.2). Therefore, universal gradient-variation regret can be applied to this problem, achieving the same *best known* bounds as Chen et al. [2024], with a *single* algorithm. Table 2 compares our results with existing ones. The following theorem presents our results formally, with the corresponding proof provided in Appendix D.2.

Table 2: Comparisons of our results with existing ones. The second column presents the regret bounds, where $\sigma_{1:T}^2$ and $\Sigma_{1:T}^2$ represent the stochastic and adversarial statistics of the SEA problem. The last column indicates whether the results can be achieved by a single algorithm (i.e., suitable in the universal setup). We achieve the same state-of-the-art guarantees as Chen et al. [2024] using one single algorithm.

| Works | Regret Bounds | | | Single Algorithm? |
|---|---|---|---|---|
| | Strongly Convex | Exp-concave | Convex | |
| Sachs et al. [2022] | $\mathcal{O}((\sigma_{\max}^2 + \Sigma_{\max}^2)\log T)$ | N/A | $\mathcal{O}(\sqrt{\sigma_{1:T}^2 + \Sigma_{1:T}^2})$ | ✗ |
| Chen et al. [2024] | $\mathcal{O}\left((\sigma_{\max}^2 + \Sigma_{\max}^2)\log\left(\frac{\sigma_{1:T}^2 + \Sigma_{1:T}^2}{\sigma_{\max}^2 + \Sigma_{\max}^2}\right)\right)$ | $\mathcal{O}(d\log(\sigma_{1:T}^2 + \Sigma_{1:T}^2))$ | $\mathcal{O}(\sqrt{\sigma_{1:T}^2 + \Sigma_{1:T}^2})$ | ✗ |
| Sachs et al. [2023] | $\mathcal{O}((\sigma_{\max}^2 + \Sigma_{\max}^2 + D^2L^2)\log^2 T)$ | N/A | $\mathcal{O}(\sqrt{T\log T})$ | ✓ |
| Yan et al. [2023] | $\mathcal{O}((\sigma_{\max}^2 + \Sigma_{\max}^2)\log(\sigma_{1:T}^2 + \Sigma_{1:T}^2))$ | $\mathcal{O}(d\log(\sigma_{1:T}^2 + \Sigma_{1:T}^2))$ | $\mathcal{O}(\sqrt{(\sigma_{1:T}^2 + \Sigma_{1:T}^2)\log(\sigma_{1:T}^2 + \Sigma_{1:T}^2)})$ | ✓ |
| **Ours** | $\mathcal{O}\left((\sigma_{\max}^2 + \Sigma_{\max}^2)\log\left(\frac{\sigma_{1:T}^2 + \Sigma_{1:T}^2}{\sigma_{\max}^2 + \Sigma_{\max}^2}\right)\right)$ | $\mathcal{O}(d\log(\sigma_{1:T}^2 + \Sigma_{1:T}^2))$ | $\mathcal{O}(\sqrt{\sigma_{1:T}^2 + \Sigma_{1:T}^2})$ | ✓ |

**Theorem 3.** *Under Assumption 1 and smoothness of $F_t(\cdot)$ for any $t \in [T]$: if $F_t(\cdot)$ is convex, Algorithm 1 achieves $\mathcal{O}(\sqrt{\sigma_{1:T}^2 + \Sigma_{1:T}^2})$; if $f_t(\cdot)$ is exp-concave, it achieves $\mathcal{O}(d\log(\sigma_{1:T}^2 + \Sigma_{1:T}^2))$; and if $F_t(\cdot)$ is strongly convex, it achieves $\mathcal{O}((\sigma_{\max}^2 + \Sigma_{\max}^2)\log((\sigma_{1:T}^2 + \Sigma_{1:T}^2)/(\sigma_{\max}^2 + \Sigma_{\max}^2)))$.*

Theorem 3 requires exp-concavity of the individual function $f_t(\cdot)$ rather than the expected function $F_t(\cdot)$. This assumption is also used by Chen et al. [2023] and common in the studies of stochastic exp-concave optimization [Mahdavi et al., 2015, Koren and Levy, 2015].

### 4.3 Discussion on Comparison with Correction-based Approach

In this part, we discuss the technical comparison with the previous correction-based approach [Yan et al., 2023]. Compared with their approach, ours is simpler and achieves the optimal universal problem-dependent regret (Theorem 1 and Theorem 2) and the best known guarantees in the SEA model (Theorem 3). Although not providing guarantees as favorable as ours, Yan et al. [2023] can control the overall algorithmic stability (i.e., $\|\mathbf{x}_t - \mathbf{x}_{t-1}\|^2$) using collaborative online ensemble [Zhao et al., 2024], which is necessary in achieving fast rates in multi-player games [Syrgkanis et al., 2015]. For example, in a min-max game $\min_{\mathbf{x}\in\mathcal{X}} \max_{\mathbf{y}\in\mathcal{Y}} \mathbf{x}^\top A\mathbf{y}$, where $A$ is a game matrix, since $A$ is unknown, the Nash equilibrium is typically computed through repeated play, i.e., player-$\mathbf{x}$ and player-$\mathbf{y}$ select $\{\mathbf{x}_t\}_{t=1}^T$ and $\{\mathbf{y}_t\}_{t=1}^T$ sequentially to approach the Nash equilibrium. For player-$\mathbf{x}$, in the $t$-th round, it suffers a loss $\mathbf{x}_t^\top A\mathbf{y}_t$ and receives the gradient $A\mathbf{y}_t$. Similarly, player-$\mathbf{y}$ suffers $-\mathbf{x}_t^\top A\mathbf{y}_t$ and receives $-A\mathbf{x}_t$. For player-$\mathbf{x}$, if it updates via OOMD, its gradient-variation regret contains $\|A\mathbf{y}_t - A\mathbf{y}_{t-1}\|^2$, which includes the stability of player-$\mathbf{y}$. In this case, to achieve fast rates, we indeed need to control the algorithm stability like $\|\mathbf{x}_t - \mathbf{x}_{t-1}\|^2$ and $\|\mathbf{y}_t - \mathbf{y}_{t-1}\|^2$. This can be done by Yan et al. [2023] while this work cannot since we do not directly control the algorithmic stability. Interested readers can refer to Appendix A.2 in Yan et al. [2023] for more details.

## 5 Conclusion

In this work, we investigate universal online learning with gradient-variation regret. We propose a simple two-layer online ensemble approach that not only achieves the optimal $\mathcal{O}(\frac{1}{\lambda}\log V_T)$, $\mathcal{O}(\frac{d}{\alpha}\log V_T)$, and $\mathcal{O}(\sqrt{V_T})$ regret simultaneously for $\lambda$-strongly convex, $\alpha$-exp-concave, and convex functions and is efficient with $\mathcal{O}(\log T)$ base learners and only 1 gradient query per round. This is done via the negative Bregman divergence term from linearization and the useful smoothness property of $\|\nabla f(\mathbf{x}) - \nabla f(\mathbf{y})\|^2 \le 2L\mathcal{D}_f(\mathbf{y}, \mathbf{x})$. We further validate the effectiveness of our approach and results by implying the optimal universal small-loss regret directly in analysis and achieving the best known results in the stochastically extended adversarial model.

Two future directions are worth investigating. The first is to reduce the number of projections to only 1 in each round [Mhammedi et al., 2019, Zhao et al., 2022, Yang et al., 2024], thereby further improving the computational efficiency. The second direction involves extending our algorithm and results to the unconstrained domain using recent advances in parameter-free online learning [Orabona and Pál, 2016, Cutkosky and Orabona, 2018, Jacobsen and Cutkosky, 2022], to broaden its applicability across a wider range of scenarios.

## Acknowledgements

This research was supported by National Science and Technology Major Project (2022ZD0114802), NSFC (62176117), and JiangsuSF (BK20220776). Peng Zhao was supported in part by the Xiaomi Foundation.

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

# A    On Smoothness Assumption

In this section, we propose a relaxation of the smoothness requirement to a slightly larger domain than the feasible domain $\mathcal{X}$, in contrast to the whole $\mathbb{R}^d$ space as in Assumption 2. The relaxed smoothness assumption is detailed in the following.

**Assumption 3** (Relaxed Smoothness). Under the condition of $\|\nabla f_t(\mathbf{x})\| \leq G$ for any $\mathbf{x} \in \mathcal{X}$ and $t \in [T]$, all online functions are $L$-smooth: $\|\nabla f_t(\mathbf{x}) - \nabla f_t(\mathbf{y})\| \leq L\|\mathbf{x} - \mathbf{y}\|$ for any $t \in [T]$ and $\mathbf{x}, \mathbf{y} \in \mathcal{X}_+$, where $\mathcal{X}_+ \triangleq \{\mathbf{x} + \mathbf{b} \mid \mathbf{x} \in \mathcal{X}, \mathbf{b} \in {}^G/_L \cdot \mathbb{B}\}$ and $\mathbb{B} \triangleq \{\mathbf{x} \mid \|\mathbf{x}\| \leq 1\}$ is a unit ball.

The domain of Assumption 3 is slightly larger than $\mathcal{X}$ — for any $\mathbf{x}_+ \in \mathcal{X}_+$, we can always find an $\mathbf{x} \in \mathcal{X}$ such that $\|\mathbf{x}_+ - \mathbf{x}\| \leq {}^G/_L$. In this work, one of the key technical contributions is to handle the empirical gradient variation via a useful smoothness property (Proposition 1). We show in the following that this condition can be satisfied by requiring only Assumption 3.

**Lemma 1.** *Under Assumption 1, for any online function $f(\cdot)$ satisfying Assumption 3, it holds that $\|\nabla f(\mathbf{x}) - \nabla f(\mathbf{y})\|^2 \leq 2L\mathcal{D}_f(\mathbf{y}, \mathbf{x})$ for any $\mathbf{x}, \mathbf{y} \in \mathcal{X}$.*

*Proof.* To begin with, we present the self-bounding property [Srebro et al., 2010], which is useful in proving our result — if a function $f : \mathbb{R}^d \mapsto \mathbb{R}$ is $L$-smooth and bounded from below, then for any $\mathbf{x} \in \mathbb{R}^d$, it holds that

$$\|\nabla f(\mathbf{x})\|^2 \leq 2L\left(f(\mathbf{x}) - \inf_{\mathbf{y} \in \mathbb{R}^d} f(\mathbf{y})\right). \tag{A.1}$$

Next, we aim to prove that if we only need (A.1) on a bounded domain $\mathcal{X}$, we require smoothness only on a slightly larger domain than $\mathcal{X}$. To see this, we delve into the proof of the self-bounding property. Specifically, for any $\mathbf{x}, \mathbf{v} \in \mathbb{R}^d$, it holds that

$$\langle -\nabla f(\mathbf{x}), \mathbf{v} \rangle - \frac{L}{2}\|\mathbf{v}\|^2 \leq f(\mathbf{x}) - f(\mathbf{x} + \mathbf{v}) \leq f(\mathbf{x}) - \inf_{\mathbf{y} \in \mathbb{R}^d} f(\mathbf{y}),$$

where the first step requires smoothness on $\mathbf{x}$ and $\mathbf{x} + \mathbf{v}$. Consequently, by taking maximization over $\mathbf{v}$, it holds that

$$f(\mathbf{x}) - \inf_{\mathbf{y} \in \mathbb{R}^d} f(\mathbf{y}) \geq \sup_{\mathbf{v} \in \mathbb{R}^d} \langle -\nabla f(\mathbf{x}), \mathbf{v} \rangle - \frac{L}{2}\|\mathbf{v}\|^2 = \frac{1}{2L}\|\nabla f(\mathbf{x})\|^2,$$

which leads to the self-bounding property (A.1) by taking $\mathbf{v} = -\frac{1}{L}\nabla f(\mathbf{x})$. The above proof is from Theorem 4.23 of Orabona [2019]. This means that for the self-bounding property, we only require the smoothness to hold for any $\mathbf{x} \in \mathcal{X}$ and $\mathbf{x} - \frac{1}{L}\nabla f(\mathbf{x})$. Under Assumption 1, this can be satisfied by requiring smoothness on a slightly larger domain than $\mathcal{X}$, namely, $\mathcal{X}_+ \triangleq \{\mathbf{x} + \mathbf{b} \mid \mathbf{x} \in \mathcal{X}, \mathbf{b} \in {}^G/_L \cdot \mathbb{B}\}$.

Now we are ready to prove the final result. To begin with, we define a surrogate function of $g(\mathbf{x}) \triangleq f(\mathbf{x}) - \langle \nabla f(\mathbf{x}_0), \mathbf{x} \rangle$ for any $\mathbf{x} \in \mathcal{X}$, where $\mathbf{x}_0 \in \mathcal{X}$. Due to the above property we have just proved, by requiring smoothness on $\mathcal{X}_+$, we have

$$\|\nabla g(\mathbf{x})\|^2 \leq 2L\left(g(\mathbf{x}) - \inf_{\mathbf{y} \in \mathbb{R}^d} g(\mathbf{y})\right).$$

Denoting by $\mathbf{y}^\star \in \arg\min_{\mathbf{y} \in \mathbb{R}^d} g(\mathbf{y})$, the above inequality equals to

$$\|\nabla f(\mathbf{x}) - \nabla f(\mathbf{x}_0)\|^2 \leq 2L\left(f(\mathbf{x}) - \langle \nabla f(\mathbf{x}_0), \mathbf{x} \rangle - f(\mathbf{y}^\star) + \langle \nabla f(\mathbf{x}_0), \mathbf{y}^\star \rangle\right)$$
$$= 2L(f(\mathbf{x}) - f(\mathbf{y}^\star) - \langle \nabla f(\mathbf{x}_0), \mathbf{x} - \mathbf{y}^\star \rangle),$$

due to the definition of $g(\cdot)$. The proof using the self-bounding property is from Theorem 2.1.5 of Nesterov [2018]. Finally, we note that $g(\cdot)$ is minimized at $\mathbf{y}^\star = \mathbf{x}_0$, leading to $\|\nabla f(\mathbf{x}) - \nabla f(\mathbf{x}_0)\|^2 \leq 2L\mathcal{D}_f(\mathbf{x}_0, \mathbf{x})$ for any $\mathbf{x}, \mathbf{x}_0 \in \mathcal{X}$, which finishes the proof. $\qquad\square$

# B    Omitted Details of Algorithm 1

In this section, we provide some omitted details of our Algorithm 1, including the losses and update rules of the base and meta learners.

**Base Learners.** To begin with, we duplicate the candidate coefficient pool (2.1) for both the exp-concave coefficient $\alpha$ and the strongly convex coefficient $\lambda$, denoted by $\mathcal{H}^{\text{exp}} \triangleq \mathcal{H}$ and $\mathcal{H}^{\text{sc}} \triangleq \mathcal{H}$. Consequently, denoting by $N^{\text{exp}} = N^{\text{sc}} \triangleq |\mathcal{H}|$ the size of candidate pool, for each $\alpha_i \in \mathcal{H}^{\text{exp}}$ and $\lambda_j \in \mathcal{H}^{\text{sc}}$, where $i \in [N^{\text{exp}}]$ and $j \in [N^{\text{sc}}]$, we define corresponding groups of base learners for optimizing exp-concave and strongly convex functions. Specifically, for $\alpha$-*exp-concave* functions, we define a group of base learners $\{\mathcal{B}_i^{\text{exp}}\}_{i\in[N^{\text{exp}}]}$, where the $i$-th base learner runs the algorithm below:

$$
\begin{aligned}
\mathbf{x}_{t,i} &= \underset{\mathbf{x}\in\mathcal{X}}{\arg\min}\left\{ \langle \nabla h_{t-1,i}^{\text{exp}}(\mathbf{x}_{t-1,i}), \mathbf{x}\rangle + \mathcal{D}_{\psi_{t,i}}(\mathbf{x}, \widehat{\mathbf{x}}_{t,i})\right\}, \\
\widehat{\mathbf{x}}_{t+1,i} &= \underset{\mathbf{x}\in\mathcal{X}}{\arg\min}\left\{ \langle \nabla h_{t,i}^{\text{exp}}(\mathbf{x}_{t,i}), \mathbf{x}\rangle + \mathcal{D}_{\psi_{t,i}}(\mathbf{x}, \widehat{\mathbf{x}}_{t,i})\right\},
\end{aligned}
\tag{B.1}
$$

where $\psi_{t,i}(\mathbf{x}) \triangleq \frac{1}{2}\mathbf{x}^\top U_{t,i}\mathbf{x}$, $U_{t,i} = (1 + \frac{\alpha_i G^2}{2})I + \frac{\alpha_i}{2}\sum_{s=1}^{t-1}\nabla h_{s,i}^{\text{exp}}(\mathbf{x}_{s,i})h_{s,i}^{\text{exp}}(\mathbf{x}_{s,i})^\top$, $\alpha_i$ is the $i$-th element in $\mathcal{H}^{\text{exp}}$, and $h_{t,i}^{\text{exp}}(\cdot)$ is a surrogate loss function for $\mathcal{B}_i^{\text{exp}}$, defined as

$$
h_{t,i}^{\text{exp}}(\mathbf{x}) \triangleq \langle \nabla f_t(\mathbf{x}_t), \mathbf{x}\rangle + \frac{\alpha_i}{4}\langle \nabla f_t(\mathbf{x}_t), \mathbf{x} - \mathbf{x}_t\rangle^2.
$$

Similarly, for $\lambda$-*strongly convex* functions, we define a group of base learners $\{\mathcal{B}_i^{\text{sc}}\}_{i\in[N^{\text{sc}}]}$, where the $i$-th base learner runs the algorithm below:

$$
\mathbf{x}_{t,i} = \Pi_\mathcal{X}[\widehat{\mathbf{x}}_{t,i} - \eta_{t,i}\nabla h_{t-1,i}^{\text{sc}}(\mathbf{x}_{t-1,i})], \quad \widehat{\mathbf{x}}_{t+1,i} = \Pi_\mathcal{X}[\widehat{\mathbf{x}}_{t,i} - \eta_{t,i}\nabla h_{t,i}^{\text{sc}}(\mathbf{x}_{t,i})],
\tag{B.2}
$$

where $\eta_{t,i} = 2/(1 + \lambda_i t)$, $\lambda_i$ is the $i$-th element in $\mathcal{H}^{\text{sc}}$, and $h_{t,i}^{\text{sc}}(\cdot)$ is a surrogate loss function for $\mathcal{B}_i^{\text{sc}}$, defined as

$$
h_{t,i}^{\text{sc}}(\mathbf{x}) \triangleq \langle \nabla f_t(\mathbf{x}_t), \mathbf{x}\rangle + \frac{\lambda_i}{4}\|\mathbf{x} - \mathbf{x}_t\|^2.
$$

For *convex* functions, we only have to define one base learner $\mathcal{B}^c$, which updates as

$$
\mathbf{x}_{t,i} = \Pi_\mathcal{X}[\widehat{\mathbf{x}}_{t,i} - \eta_{t,i}\nabla f_{t-1}(\mathbf{x}_{t-1})], \quad \widehat{\mathbf{x}}_{t+1,i} = \Pi_\mathcal{X}[\widehat{\mathbf{x}}_{t,i} - \eta_{t,i}\nabla f_t(\mathbf{x}_t)],
\tag{B.3}
$$

where $\eta_{t,i} = \min\{D/\sqrt{1 + \sum_{s=2}^{t-1}\|\nabla f_t(\mathbf{x}_t) - \nabla f_{t-1}(\mathbf{x}_{t-1})\|^2}, 1\}$. Finally, we conclude the configurations of base learners. Specifically, we deploy

$$
\{\mathcal{B}_i\}_{i\in[N]} \triangleq \{\mathcal{B}_i^{\text{exp}}\}_{i\in[N^{\text{exp}}]} \cup \{\mathcal{B}_i^{\text{sc}}\}_{i\in[N^{\text{sc}}]} \cup \{\mathcal{B}^c\}, \text{ where } N \triangleq N^{\text{exp}} + N^{\text{sc}} + 1,
\tag{B.4}
$$

as the total set of base learners.

**Meta Learner.** The meta learner simply runs OPTIMISTIC-ADAPT-ML-PROD [Wei et al., 2016], which updates as follows:

$$
\begin{aligned}
p_{t+1,i} &\propto \varepsilon_{t,i} \cdot \exp(\varepsilon_{t,i}m_{t+1,i}) \cdot W_{t,i}, \\
W_{t,i} &= \left(W_{t-1,i} \cdot \exp\left(\varepsilon_{t-1,i}r_{t,i} - \varepsilon_{t-1,i}^2(r_{t,i} - m_{t,i})\right)\right)^{\frac{\varepsilon_{t,i}}{\varepsilon_{t-1,i}}},
\end{aligned}
\tag{B.5}
$$

where $\ell_{t,i} \triangleq \langle \nabla f_t(\mathbf{x}_t), \mathbf{x}_{t,i}\rangle$ is the loss of the $i$-th dimension, $r_{t,i} = \langle \boldsymbol{\ell}_t, \boldsymbol{p}_t\rangle - \ell_{t,i}$ represents the instantaneous regret, $m_{t,i} = \langle \nabla f_{t-1}(\mathbf{x}_{t-1}), \mathbf{x}_t - \mathbf{x}_{t,i}\rangle$ for the index $i$ indicating $\mathcal{B}^c$ and $m_{t,i} = 0$ for indexes indicating $\mathcal{B}^{\text{exp}}$ and $\mathcal{B}^{\text{sc}}$. The learning rate $\varepsilon_{t,i}$ is chosen as

$$
\varepsilon_{t,i} = \min\left\{\frac{1}{8}, \sqrt{\frac{\ln N}{\sum_{s\in[t]}(r_{s,i} - m_{s,i})^2}}\right\}.
\tag{B.6}
$$

# C Proof for Section 3

In this section, we provide the proof of Theorem 1, our main theoretical result for the optimal universal gradient-variation regret.

*Proof.* We first give different decompositions of the regret for different curvature types, then analyze the meta and base regret, and finally combine them to achieve the regret bound. For simplicity, we define $\mathbf{g}_t \triangleq \nabla f_t(\mathbf{x}_t)$ and

$$
\bar{V}_T \triangleq \sum_{t=2}^{T}\|\nabla f_t(\mathbf{x}_t) - \nabla f_{t-1}(\mathbf{x}_{t-1})\|^2
$$

for short. Using the analysis in Section 3.2, the empirical gradient variation can bounded as

$$\bar{V}_T \leq 3 \sum_{t=2}^{T} \|\nabla f_t(\mathbf{x}_t) - \nabla f_t(\mathbf{x}^\star)\|^2 + 3 \sum_{t=2}^{T} \|\nabla f_t(\mathbf{x}^\star) - \nabla f_{t-1}(\mathbf{x}^\star)\|^2$$

$$+ 3 \sum_{t=2}^{T} \|\nabla f_{t-1}(\mathbf{x}^\star) - \nabla f_{t-1}(\mathbf{x}_{t-1})\|^2 \leq 6L \sum_{t=2}^{T} \mathcal{D}_{f_t}(\mathbf{x}^\star, \mathbf{x}_t) + 3V_T + 6L \sum_{t=2}^{T} \mathcal{D}_{f_{t-1}}(\mathbf{x}^\star, \mathbf{x}_{t-1})$$

$$\leq 3V_T + 12L \sum_{t=1}^{T} \mathcal{D}_{f_t}(\mathbf{x}^\star, \mathbf{x}_t). \tag{C.1}$$

**Regret Decomposition.** Denoting by $\mathbf{x}^\star \in \arg\min_{\mathbf{x} \in \mathcal{X}} \sum_{t \in [T]} f_t(\mathbf{x})$, for *convex* functions, we decompose the regret as

$$\text{REG}_T = \sum_{t=1}^{T} \langle \mathbf{g}_t, \mathbf{x}_t - \mathbf{x}^\star \rangle - \sum_{t=1}^{T} \mathcal{D}_{f_t}(\mathbf{x}^\star, \mathbf{x}_t) \tag{by (3.3)}$$

$$= \underbrace{\sum_{t=1}^{T} \langle \mathbf{g}_t, \mathbf{x}_t - \mathbf{x}_{t,i^\star} \rangle}_{\text{META-REG}} + \underbrace{\sum_{t=1}^{T} h_{t,i^\star}^{\text{c}}(\mathbf{x}_{t,i^\star}) - \sum_{t=1}^{T} h_{t,i^\star}^{\text{c}}(\mathbf{x}^\star)}_{\text{BASE-REG}} - \sum_{t=1}^{T} \mathcal{D}_{f_t}(\mathbf{x}^\star, \mathbf{x}_t), \tag{C.2}$$

where $h_{t,i}^{\text{c}}(\mathbf{x}) \triangleq \langle \mathbf{g}_t, \mathbf{x} \rangle$.

For $\alpha$-*exp-concave* functions, we decompose the regret as

$$\text{REG}_T = \sum_{t=1}^{T} \langle \mathbf{g}_t, \mathbf{x}_t - \mathbf{x}^\star \rangle - \frac{1}{2} \sum_{t=1}^{T} \mathcal{D}_{f_t}(\mathbf{x}^\star, \mathbf{x}_t) - \frac{1}{2} \sum_{t=1}^{T} \mathcal{D}_{f_t}(\mathbf{x}^\star, \mathbf{x}_t) \tag{by (3.3)}$$

$$\leq \sum_{t=1}^{T} \langle \mathbf{g}_t, \mathbf{x}_t - \mathbf{x}^\star \rangle - \frac{\alpha}{4} \sum_{t=1}^{T} \langle \mathbf{g}_t, \mathbf{x}_t - \mathbf{x}^\star \rangle^2 - \frac{1}{2} \sum_{t=1}^{T} \mathcal{D}_{f_t}(\mathbf{x}^\star, \mathbf{x}_t)$$

$$\leq \underbrace{\sum_{t=1}^{T} \langle \mathbf{g}_t, \mathbf{x}_t - \mathbf{x}_{t,i^\star} \rangle - \frac{\alpha_{i^\star}}{4} \sum_{t=1}^{T} \langle \mathbf{g}_t, \mathbf{x}_t - \mathbf{x}_{t,i^\star} \rangle^2}_{\text{META-REG}} \tag{by $\alpha_{i^\star} \leq \alpha \leq 2\alpha_{i^\star}$}$$

$$+ \underbrace{\sum_{t=1}^{T} h_{t,i^\star}^{\text{exp}}(\mathbf{x}_{t,i^\star}) - \sum_{t=1}^{T} h_{t,i^\star}^{\text{exp}}(\mathbf{x}^\star)}_{\text{BASE-REG}} - \frac{1}{2} \sum_{t=1}^{T} \mathcal{D}_{f_t}(\mathbf{x}^\star, \mathbf{x}_t), \tag{C.3}$$

where the second step is due to the definitions of exp-concavity and Bregman divergence and the last step is due to the definition of the surrogate function $h_{t,i}^{\text{exp}}(\mathbf{x}) \triangleq \langle \mathbf{g}_t, \mathbf{x} \rangle + \frac{\alpha_i}{4} \langle \nabla f_t(\mathbf{x}_t), \mathbf{x} - \mathbf{x}_t \rangle^2$, where $\alpha_i \in \mathcal{H}$, defined in (2.1).

For $\lambda$-*strongly convex* functions, following the similar decomposition in the exp-concavity case,

$$\text{REG}_T \leq \underbrace{\sum_{t=1}^{T} \langle \mathbf{g}_t, \mathbf{x}_t - \mathbf{x}_{t,i^\star} \rangle - \frac{\lambda_{i^\star}}{4} \sum_{t=1}^{T} \|\mathbf{x}_t - \mathbf{x}_{t,i^\star}\|^2}_{\text{META-REG}} \tag{by $\lambda_{i^\star} \leq \lambda \leq 2\lambda_{i^\star}$}$$

$$+ \underbrace{\sum_{t=1}^{T} h_{t,i^\star}^{\text{sc}}(\mathbf{x}_{t,i^\star}) - \sum_{t=1}^{T} h_{t,i^\star}^{\text{sc}}(\mathbf{x}^\star)}_{\text{BASE-REG}} - \frac{1}{2} \sum_{t=1}^{T} \mathcal{D}_{f_t}(\mathbf{x}^\star, \mathbf{x}_t), \tag{C.4}$$

due to the definition of the surrogate $h_{t,i}^{\text{sc}}(\mathbf{x}) \triangleq \langle \mathbf{g}_t, \mathbf{x} \rangle + \frac{\lambda_i}{4} \|\mathbf{x} - \mathbf{x}_t\|^2$, where $\lambda_i \in \mathcal{H}$ in (2.1).

**Meta Regret Analysis.** We adopt OPTIMISTIC-ADAPT-ML-PROD [Wei et al., 2016] as the meta learner, and present its regret analysis below for self-containedness.

**Lemma 2** (Theorem 3.4 of Wei et al. [2016]). *Denoting by $\boldsymbol{p}_t$ the weights of the algorithm, $\boldsymbol{\ell}_t$ the loss vector, and $m_{t,i}$ the optimism, by choosing the learning rate optimally as* (B.6)*, the regret of* OPTIMISTIC-ADAPT-ML-PROD (B.5) *with respect to any expert $i \in [N]$ satisfies*

$$\sum_{t=1}^{T} \langle \boldsymbol{\ell}_t, \boldsymbol{p}_t - \boldsymbol{e}_i \rangle \le C_0 \sqrt{1 + \sum_{t=1}^{T} (r_{t,i} - m_{t,i})^2} + C_1,$$

*where $r_{t,i} = \langle \boldsymbol{\ell}_t, \boldsymbol{p}_t - \boldsymbol{e}_i \rangle$, $\boldsymbol{e}_i$ denotes the $i$-th standard basis vector, $C_0 = \sqrt{\ln N} + \ln(1 + \frac{N}{e}(1 + \ln(T+1)))/\sqrt{\ln N}$, and $C_1 = \frac{1}{4}(\ln N + \ln(1 + \frac{N}{e}(1 + \ln(T+1)))) + 2\sqrt{\ln N} + 16\ln N$.*

Here we adopt $\ell_{t,i} = \langle \mathbf{g}_t, \mathbf{x}_{t,i} \rangle$ such that $\langle \boldsymbol{\ell}_t, \boldsymbol{p}_t - \boldsymbol{e}_i \rangle = \langle \mathbf{g}_t, \mathbf{x}_t - \mathbf{x}_{t,i} \rangle$. Besides, since the number of base learners $N = \mathcal{O}(\log T)$ as explained in Section 2, the constants $C_0$ and $C_1$ are in the order of $\mathcal{O}(\log \log T)$ and can be treated as ignorable constants, following previous convention [Luo and Schapire, 2015, Gaillard et al., 2014].

For *convex* functions, we choose the optimism as $m_{t,i} = \langle \mathbf{g}_{t-1}, \mathbf{x}_t - \mathbf{x}_{t,i} \rangle$ for the index $i$ indicating the convex base learner. As explained in Section 3.1, although $\mathbf{x}_t$ is unknown for now, we only require the scalar value of $\langle \mathbf{g}_{t-1}, \mathbf{x}_t \rangle$. Denoting by $z = \langle \mathbf{g}_{t-1}, \mathbf{x}_t \rangle$, it actually forms a fixed-point problem of $z = \langle \mathbf{g}_{t-1}, \mathbf{x}_t(z) \rangle$, where $\mathbf{x}_t$ is a function of $z$ since $\mathbf{x}_t$ depends on $p_{t,i}$, $p_{t,i}$ relies on $m_{t,i}$, and $m_{t,i}$ depends on $z$. Such a one-dimensional fixed-point problem can be solved with an $\mathcal{O}(1/T)$ approximation error through $\mathcal{O}(\log T)$ binary searches, and aggregating the approximate error over the whole time horizon will only incur an additive constant to the final regret. As a result, such an optimism setup is valid. Consequently, the meta regret in (C.2) can be bounded as

$$\text{META-REG} \le C_0 \sqrt{1 + \sum_{t=1}^{T} \langle \mathbf{g}_t - \mathbf{g}_{t-1}, \mathbf{x}_t - \mathbf{x}_{t,i^\star} \rangle^2} + C_1 \qquad \text{(by Lemma 2)}$$

$$\le C_0 \sqrt{1 + D^2 \bar{V}_T} + C_1 \le C_0 \sqrt{1 + 3D^2 V_T + 12LD^2 \sum_{t=1}^{T} \mathcal{D}_{f_t}(\mathbf{x}^\star, \mathbf{x}_t)} + C_1 \qquad \text{(by (C.1))}$$

$$\le \mathcal{O}(\sqrt{V_T}) + C_0 \sqrt{12LD^2 \sum_{t=1}^{T} \mathcal{D}_{f_t}(\mathbf{x}^\star, \mathbf{x}_t)} \le \mathcal{O}(\sqrt{V_T}) + \mathcal{O}(C_2) + \frac{C_0}{2C_2} \sum_{t=1}^{T} \mathcal{D}_{f_t}(\mathbf{x}^\star, \mathbf{x}_t),$$

where the second step adopts Assumption 1, the fourth step uses $\sqrt{a+b} \le \sqrt{a} + \sqrt{b}$ for any $a, b \ge 0$, the last step uses AM-GM inequality: $\sqrt{ab} \le \frac{ax}{2} + \frac{b}{2x}$ for any $a, b, x > 0$. Note that $C_2$ is used to ensure the positive Bregman divergence term to be canceled and will be specified in the end.

For *exp-concave* functions, we choose the optimism as $m_{t,i} = 0$ for indexes $i$ indicating exp-concave base learners. By Lemma 2, the meta regret in (C.3) can be bounded as

$$\text{META-REG} \le C_0 \sqrt{1 + \sum_{t=1}^{T} \langle \mathbf{g}_t, \mathbf{x}_t - \mathbf{x}_{t,i^\star} \rangle^2 - \frac{\alpha_{i^\star}}{4} \sum_{t=1}^{T} \langle \mathbf{g}_t, \mathbf{x}_t - \mathbf{x}_{t,i^\star} \rangle^2} + C_1$$

$$\le \mathcal{O}(C_3) + \left( \frac{C_0}{2C_3} - \frac{\alpha_{i^\star}}{4} \right) \sum_{t=1}^{T} \langle \mathbf{g}_t, \mathbf{x}_t - \mathbf{x}_{t,i^\star} \rangle^2, \qquad \text{(C.5)}$$

where the last step omits the ignorable additive $C_0$ or $C_1$ terms and is due to AM-GM inequality. $C_2$ is a constant to be specified.

For *strongly convex* functions, we choose the optimism $m_{t,i} = 0$ for indexes $i$ indicating strongly convex base learners. By Lemma 2, the meta regret in (C.4) can be bounded as

$$\text{META-REG} \le C_0 \sqrt{1 + \sum_{t=1}^{T} \langle \mathbf{g}_t, \mathbf{x}_t - \mathbf{x}_{t,i^\star} \rangle^2 - \frac{\lambda_{i^\star}}{4} \sum_{t=1}^{T} \|\mathbf{x}_t - \mathbf{x}_{t,i^\star}\|^2} + C_1 \qquad \text{(by Lemma 2)}$$

$$\le C_0 \sqrt{1 + D^2 \sum_{t=1}^{T} \|\mathbf{x}_t - \mathbf{x}_{t,i^\star}\|^2 - \frac{\lambda_{i^\star}}{4} \sum_{t=1}^{T} \|\mathbf{x}_t - \mathbf{x}_{t,i^\star}\|^2} + C_1 \qquad \text{(by Assumption 1)}$$

$$\leq \mathcal{O}(C_4) + \left( \frac{C_0 D^2}{2C_4} - \frac{\lambda_{i^\star}}{4} \right) \sum_{t=1}^{T} \|\mathbf{x}_t - \mathbf{x}_{t,i^\star}\|^2, \tag{C.6}$$

where the last step omits the ignorable additive $C_0$ or $C_1$ terms and is due to AM-GM inequality. $C_4$ is a constant to be specified.

**Base Regret Analysis.** For *convex* functions, when using the update rule (B.3), due to the standard analysis of OOMD for convex functions (e.g., Lemma 10 of Yan et al. [2023]), it holds that

$$\textsc{Base-Reg} \leq 5D \sqrt{1 + \sum_{t=2}^{T} \|\nabla h_{t,i^\star}^{\mathrm{c}}(\mathbf{x}_{t,i^\star}) - \nabla h_{t-1,i^\star}^{\mathrm{c}}(\mathbf{x}_{t-1,i^\star})\|^2} + \mathcal{O}(1)$$

$$= 5D\sqrt{1 + \bar{V}_T} + \mathcal{O}(1) \leq \mathcal{O}(\sqrt{V_T}) + 5D \sqrt{12L \sum_{t=1}^{T} \mathcal{D}_{f_t}(\mathbf{x}^\star, \mathbf{x}_t)} \qquad \text{(by (C.1))}$$

$$\leq \mathcal{O}(\sqrt{V_T}) + \mathcal{O}(C_5) + \frac{5D}{2C_5} \sum_{t=1}^{T} \mathcal{D}_{f_t}(\mathbf{x}^\star, \mathbf{x}_t),$$

where the second step is due to the property of the surrogate function: $\nabla h_{t,i}^{\mathrm{c}}(\mathbf{x}_{t,i}) = \mathbf{g}_t$, and the last step uses AM-GM inequality. $C_5$ is a constant to be specified.

For *exp-concave* functions, when using the update rule (B.1), due to the standard analysis of OOMD for exp-concave functions (e.g., Lemma 11 of Yan et al. [2023]), the base regret can be bounded as

$$\textsc{Base-Reg} \leq \frac{16d}{\alpha_{i^\star}} \ln \left( 1 + \frac{\alpha_{i^\star}}{8d} \sum_{t=2}^{T} \left\| \nabla h_{t,i^\star}^{\exp}(\mathbf{x}_{t,i^\star}) - \nabla h_{t-1,i^\star}^{\exp}(\mathbf{x}_{t-1,i^\star}) \right\|^2 \right) + \mathcal{O}(1). \tag{C.7}$$

Next, we analyze the empirical gradient variation defined on the surrogate function $h_{t,i}^{\exp}(\cdot)$:

$$\sum_{t=2}^{T} \left\| \nabla h_{t,i^\star}^{\exp}(\mathbf{x}_{t,i^\star}) - \nabla h_{t-1,i^\star}^{\exp}(\mathbf{x}_{t-1,i^\star}) \right\|^2$$

$$= \sum_{t=2}^{T} \left\| \mathbf{g}_t + \frac{\alpha_{i^\star}}{2} \mathbf{g}_t \langle \mathbf{g}_t, \mathbf{x}_t - \mathbf{x}_{t,i^\star} \rangle - \mathbf{g}_{t-1} - \frac{\alpha_{i^\star}}{2} \mathbf{g}_{t-1} \langle \mathbf{g}_{t-1}, \mathbf{x}_{t-1} - \mathbf{x}_{t-1,i^\star} \rangle \right\|^2$$

$$\leq 3\bar{V}_T + 3 \sum_{t=2}^{T} \left\| \frac{\alpha_{i^\star}}{2} \mathbf{g}_t \langle \mathbf{g}_t, \mathbf{x}_t - \mathbf{x}_{t,i^\star} \rangle \right\|^2 + 3 \sum_{t=2}^{T} \left\| \frac{\alpha_{i^\star}}{2} \mathbf{g}_{t-1} \langle \mathbf{g}_{t-1}, \mathbf{x}_{t-1} - \mathbf{x}_{t-1,i^\star} \rangle \right\|^2$$

$$\leq 3\bar{V}_T + 6 \sum_{t=1}^{T} \left\| \frac{\alpha_{i^\star}}{2} \mathbf{g}_t \langle \mathbf{g}_t, \mathbf{x}_t - \mathbf{x}_{t,i^\star} \rangle \right\|^2 \tag{C.8}$$

$$\leq 9V_T + 36L \sum_{t=1}^{T} \mathcal{D}_{f_t}(\mathbf{x}^\star, \mathbf{x}_t) + 2\alpha_{i^\star}^2 G^2 \sum_{t=1}^{T} \langle \mathbf{g}_t, \mathbf{x}_t - \mathbf{x}_{t,i^\star} \rangle^2, \qquad \text{(by (C.1) and Assumption 1)}$$

where the first step is due to the property of the surrogate function: $\nabla h_{t,i}^{\exp}(\mathbf{x}_{t,i}) = \mathbf{g}_t + \frac{\alpha_i}{2} \mathbf{g}_t \langle \mathbf{g}_t, \mathbf{x}_t - \mathbf{x}_{t,i} \rangle$, the second step is by the Cauchy-Schwarz inequality: $(a + b + c)^2 \leq 3(a^2 + b^2 + c^2)$ for any $a, b, c \in \mathbb{R}$. Plugging the surrogate's empirical gradient variation back to the base regret, we obtain

$$\textsc{Base-Reg} \leq \frac{16d}{\alpha_{i^\star}} \ln \left( 1 + \frac{9\alpha_{i^\star}}{8d} V_T + \frac{9\alpha_{i^\star} L}{2d} \sum_{t=1}^{T} \mathcal{D}_{f_t}(\mathbf{x}^\star, \mathbf{x}_t) + \frac{\alpha_{i^\star}^3 G^2}{4d} \sum_{t=1}^{T} \langle \mathbf{g}_t, \mathbf{x}_t - \mathbf{x}_{t,i^\star} \rangle^2 \right)$$

$$\leq \mathcal{O}\left( \frac{d}{\alpha} \ln(C_6 V_T) \right) + \frac{16d}{C_6 \alpha_{i^\star}} \left( \frac{9\alpha_{i^\star} L}{2d} \sum_{t=1}^{T} \mathcal{D}_{f_t}(\mathbf{x}^\star, \mathbf{x}_t) + \frac{\alpha_{i^\star}^3 G^2}{4d} \sum_{t=1}^{T} \langle \mathbf{g}_t, \mathbf{x}_t - \mathbf{x}_{t,i^\star} \rangle^2 \right)$$

$$\leq \mathcal{O}\left( \frac{d}{\alpha} \ln V_T \right) + \frac{72L}{C_6} \sum_{t=1}^{T} \mathcal{D}_{f_t}(\mathbf{x}^\star, \mathbf{x}_t) + \frac{4G^2}{C_6} \sum_{t=1}^{T} \langle \mathbf{g}_t, \mathbf{x}_t - \mathbf{x}_{t,i^\star} \rangle^2 + \mathcal{O}(\ln C_6).$$

The second step requires $C_6 \geq 1$ by Lemma 5 and uses the property of the best base learner, i.e., $\alpha_{i^\star} \leq \alpha \leq 2\alpha_{i^\star}$. The last step is because of $\alpha_i \leq 1$.

For *strongly convex* functions, when using the update rule (B.2), due to the analysis of OOMD for strongly convex functions (e.g., Lemma 12 of Yan et al. [2023]), the base regret can be bounded as

$$\text{BASE-REG} \leq \frac{16G^2}{\lambda_{i^\star}} \ln\left(1 + \lambda_{i^\star} \sum_{t=2}^{T} \left\|\nabla h_{t,i^\star}^{\text{sc}}(\mathbf{x}_{t,i^\star}) - \nabla h_{t-1,i^\star}^{\text{sc}}(\mathbf{x}_{t-1,i^\star})\right\|^2\right) + \mathcal{O}(1). \quad \text{(C.9)}$$

Next, we analyze the empirical gradient variation defined on the surrogate function $h_{t,i}^{\text{sc}}(\cdot)$:

$$\sum_{t=2}^{T} \left\|\nabla h_{t,i^\star}^{\text{sc}}(\mathbf{x}_{t,i^\star}) - \nabla h_{t-1,i^\star}^{\text{sc}}(\mathbf{x}_{t-1,i^\star})\right\|^2$$

$$= \sum_{t=2}^{T} \left\|\mathbf{g}_t + \frac{\lambda_{i^\star}}{2}(\mathbf{x}_{t,i^\star} - \mathbf{x}_t) - \mathbf{g}_{t-1} - \frac{\lambda_{i^\star}}{2}(\mathbf{x}_{t-1,i^\star} - \mathbf{x}_{t-1})\right\|^2$$

$$\leq 3\bar{V}_T + 3\sum_{t=2}^{T}\left\|\frac{\lambda_{i^\star}}{2}(\mathbf{x}_{t,i^\star} - \mathbf{x}_t)\right\|^2 + 3\sum_{t=2}^{T}\left\|\frac{\lambda_{i^\star}}{2}(\mathbf{x}_{t-1,i^\star} - \mathbf{x}_{t-1})\right\|^2 \quad \text{(C.10)}$$

$$\leq 9V_T + 36L\sum_{t=1}^{T}\mathcal{D}_{f_t}(\mathbf{x}^\star, \mathbf{x}_t) + 2\lambda_{i^\star}^2\sum_{t=1}^{T}\|\mathbf{x}_{t,i^\star} - \mathbf{x}_t\|^2, \quad \text{(by (C.1))}$$

where the first step is due to the property of the surrogate: $\nabla h_{t,i}^{\text{sc}}(\mathbf{x}_{t,i}) = \mathbf{g}_t + \frac{\lambda_i}{2}(\mathbf{x}_{t,i} - \mathbf{x}_t)$, and the second step is due to the Cauchy-Schwarz inequality. Plugging the surrogate's empirical gradient variation back to the base regret, we obtain

$$\text{BASE-REG} \leq \frac{16G^2}{\lambda_{i^\star}} \ln\left(1 + 9\lambda_{i^\star} V_T + 36L\lambda_{i^\star}\sum_{t=1}^{T}\mathcal{D}_{f_t}(\mathbf{x}^\star, \mathbf{x}_t) + 2\lambda_{i^\star}^3\sum_{t=1}^{T}\|\mathbf{x}_{t,i^\star} - \mathbf{x}_t\|^2\right)$$

$$\leq \mathcal{O}\left(\frac{1}{\lambda}\ln(C_7 V_T)\right) + \frac{16G^2}{C_7\lambda_{i^\star}}\left(36L\lambda_{i^\star}\sum_{t=1}^{T}\mathcal{D}_{f_t}(\mathbf{x}^\star, \mathbf{x}_t) + 2\lambda_{i^\star}^3\sum_{t=1}^{T}\|\mathbf{x}_{t,i^\star} - \mathbf{x}_t\|^2\right)$$

$$\leq \mathcal{O}\left(\frac{1}{\lambda}\ln V_T\right) + \frac{576G^2 L}{C_7}\sum_{t=1}^{T}\mathcal{D}_{f_t}(\mathbf{x}^\star, \mathbf{x}_t) + \frac{32G^2}{C_7}\sum_{t=1}^{T}\|\mathbf{x}_{t,i^\star} - \mathbf{x}_t\|^2 + \mathcal{O}(\ln C_7),$$

where the second step requires $C_7 \geq 1$ by Lemma 5 and uses the property of the best base learner, i.e., $\lambda_{i^\star} \leq \lambda \leq 2\lambda_{i^\star}$. The last step is due to $\lambda_i \leq 1$.

**Regret Analysis.** For *convex* functions, by combining the meta and base regret, it holds that

$$\text{REG}_T \leq \mathcal{O}(\sqrt{V_T}) + \mathcal{O}(C_2 + C_5) + \left(\frac{C_0}{2C_2} + \frac{5D}{2C_5} - 1\right)\sum_{t=1}^{T}\mathcal{D}_{f_t}(\mathbf{x}^\star, \mathbf{x}_t) \leq \mathcal{O}(\sqrt{V_T}),$$

by choosing $C_2 = C_0$ and $C_5 = 5D$.

For *exp-concave* functions, by combining the meta and base regret, it holds that

$$\text{REG}_T \leq \mathcal{O}\left(\frac{d}{\alpha}\ln V_T\right) + \mathcal{O}(C_3 + \ln C_6) + \left(\frac{C_0}{2C_3} + \frac{4G^2}{C_6} - \frac{\alpha_{i^\star}}{4}\right)\sum_{t=1}^{T}\langle\mathbf{g}_t, \mathbf{x}_t - \mathbf{x}_{t,i^\star}\rangle^2$$

$$+ \left(\frac{72L}{C_6} - \frac{1}{2}\right)\sum_{t=1}^{T}\mathcal{D}_{f_t}(\mathbf{x}^\star, \mathbf{x}_t) \leq \mathcal{O}\left(\frac{d}{\alpha}\ln V_T\right),$$

by choosing $C_6 = \max\{1, 144L, \frac{32G^2}{\alpha_{i^\star}}\}$ and $C_3 = \frac{4C_0}{\alpha_{i^\star}}$. Note that such a parameter configuration will only add an $\mathcal{O}(1/\alpha)$ factor to the final regret bound, which can be absorbed.

For *strongly convex* functions, by combining the meta and base regret, it holds that

$$\text{REG}_T \leq \mathcal{O}\left(\frac{1}{\lambda}\ln V_T\right) + \mathcal{O}(C_4 + \ln C_7) + \left(\frac{C_0 D^2}{2C_4} + \frac{32G^2}{C_7} - \frac{\lambda_{i^\star}}{4}\right)\sum_{t=1}^{T}\|\mathbf{x}_t - \mathbf{x}_{t,i^\star}\|^2$$

$$+ \left( \frac{576G^2L}{C_7} - \frac{1}{2} \right) \sum_{t=1}^{T} \mathcal{D}_{f_t}(\mathbf{x}^\star, \mathbf{x}_t) \leq \mathcal{O}\left( \frac{1}{\lambda} \ln V_T \right),$$

by choosing $C_7 = \max\{1, 1152G^2L, \frac{256G^2}{\lambda_{i^\star}}\}$ and $C_4 = \frac{4C_0D^2}{\lambda_{i^\star}}$. Note that such a parameter configuration will only add an $\mathcal{O}(1/\lambda)$ factor to the final regret bound, which can be absorbed.

Note that the constants $C_2, C_3, C_4, C_5, C_6, C_7$ only exist in analysis and thus can be chosen arbitrarily, finishing the proof. □

## D   Proofs for Section 4

In this section, we provide proofs for Section 4, including Theorem 2 and Theorem 3.

### D.1   Proof of Theorem 2

*Proof.* To begin with, we give a different decomposition for the empirical gradient variation $\bar{V}_T$:

$$\bar{V}_T \leq 2\sum_{t=2}^{T} \|\mathbf{g}_t\|^2 + 2\sum_{t=2}^{T} \|\mathbf{g}_{t-1}\|^2 \leq 4\sum_{t=1}^{T} \|\mathbf{g}_t\|^2 \leq 16L\sum_{t=1}^{T} f_t(\mathbf{x}_t), \qquad \text{(D.1)}$$

where the last step is by the self-bounding property (A.1) for non-negative functions. For simplicity, we denote by $\bar{F}_T \triangleq \sum_{t=1}^{T} f_t(\mathbf{x}_t)$.

The regret decomposition is the same as that in the proof of Theorem 1 (i.e., (C.2), (C.3), and (C.4)), and thus omitted here. In the following, we analyze the meta and base regret, and combine them for the final regret bounds.

**Meta Regret Analysis.**   Our Algorithm 1 achieves the small-loss bounds without modifying the algorithm. As a result, the meta algorithm and the corresponding step size and optimism configurations are the same as that in the proof of Theorem 1.

For *convex* functions, by choosing the optimism as $m_{t,i} = \langle \mathbf{g}_{t-1}, \mathbf{x}_t - \mathbf{x}_{t,i} \rangle$ for the index $i$ indicating the base learner for the convex case, the meta regret in (C.2) can be bounded as

$$\text{META-REG} \leq C_0 \sqrt{1 + \sum_{t=1}^{T} \langle \mathbf{g}_t - \mathbf{g}_{t-1}, \mathbf{x}_t - \mathbf{x}_{t,i^\star} \rangle^2} + C_1 \qquad \text{(by Lemma 2)}$$

$$\leq C_0 \sqrt{1 + D^2\bar{V}_T} + C_1 \leq C_0\sqrt{1 + 16D^2L\bar{F}_T} + C_1. \qquad \text{(by Assumption 1 and (D.1))}$$

For *exp-concave* functions, we choose the optimism as $m_{t,i} = 0$ for indexes $i$ indicating the exp-concave base learners. The meta regret is bounded in the same way as (C.5).

For *strongly convex* functions, we choose the optimism as $m_{t,i} = 0$ for indexes $i$ indicating the strongly convex base learners. The meta regret is bounded in the same way as (C.6).

**Base Regret Analysis.**   For *convex* functions, using the same base algorithms as in the proof of Theorem 1, the base regret can be bounded as

$$\text{BASE-REG} \leq 5D\sqrt{1 + \bar{V}_T} + \mathcal{O}(1) \leq 5D\sqrt{1 + 16L\bar{F}_T} + \mathcal{O}(1).$$

For *exp-concave* functions, using the same base algorithms as in the proof of Theorem 1, the base regret can be bounded by (C.7). Following (C.8), the empirical gradient variation defined on the surrogate function $h_{t,i}^{\exp}(\cdot)$ can be bounded as

$$\sum_{t=2}^{T} \left\| \nabla h_{t,i^\star}^{\exp}(\mathbf{x}_{t,i^\star}) - \nabla h_{t-1,i^\star}^{\exp}(\mathbf{x}_{t-1,i^\star}) \right\|^2 \leq 3\bar{V}_T + 6\sum_{t=1}^{T} \left\| \frac{\alpha_{i^\star}}{2}\mathbf{g}_t\langle \mathbf{g}_t, \mathbf{x}_t - \mathbf{x}_{t,i^\star} \rangle \right\|^2$$

$$\leq 48L\bar{F}_T + 2\alpha_{i^\star}^2 G^2 \sum_{t=1}^{T} \langle \mathbf{g}_t, \mathbf{x}_t - \mathbf{x}_{t,i^\star} \rangle^2. \qquad \text{(by Assumption 1 and (D.1))}$$

Plugging the surrogate's empirical gradient variation back to the base regret, we obtain

$$\text{BASE-REG} \leq \frac{16d}{\alpha_{i^\star}} \ln\left(1 + \frac{6L\alpha_{i^\star}}{d}\bar{F}_T + \frac{\alpha_{i^\star}^3 G^2}{4d} \sum_{t=1}^T \langle \mathbf{g}_t, \mathbf{x}_t - \mathbf{x}_{t,i^\star}\rangle^2\right) + \mathcal{O}(1)$$

$$\leq \frac{16d}{\alpha_{i^\star}} \ln\left(C_8\left(1 + \frac{6L\alpha_{i^\star}}{d}\bar{F}_T\right)\right) + \frac{16d}{C_8\alpha_{i^\star}}\left(\frac{\alpha_{i^\star}^3 G^2}{4d}\sum_{t=1}^T \langle \mathbf{g}_t, \mathbf{x}_t - \mathbf{x}_{t,i^\star}\rangle^2\right)$$

$$\leq \frac{32d}{\alpha}\ln\left(1 + \frac{6L}{d}\bar{F}_T\right) + \frac{4G^2}{C_8}\sum_{t=1}^T \langle \mathbf{g}_t, \mathbf{x}_t - \mathbf{x}_{t,i^\star}\rangle^2 + \mathcal{O}(\ln C_8),$$

where the second step requires $C_8 \geq 1$ by Lemma 5 and uses the property of the best base learner, i.e., $\alpha_{i^\star} \leq \alpha \leq 2\alpha_{i^\star}$. The last step is due to $\alpha_i \leq 1$.

For *strongly convex* functions, using the same base algorithms as in the proof of Theorem 1, the base regret can be bounded by (C.9). Following (C.10), the empirical gradient variation defined on the surrogate function $h_{t,i}^{\text{sc}}(\cdot)$ can be bounded as

$$\sum_{t=2}^T \left\|\nabla h_{t,i^\star}^{\text{sc}}(\mathbf{x}_{t,i^\star}) - \nabla h_{t-1,i^\star}^{\text{sc}}(\mathbf{x}_{t-1,i^\star})\right\|^2 \leq 3\bar{V}_T + 6\sum_{t=1}^T \left\|\frac{\lambda_{i^\star}}{2}(\mathbf{x}_{t,i^\star} - \mathbf{x}_t)\right\|^2$$

$$\leq 48L\bar{F}_T + 2\lambda_{i^\star}^2 \sum_{t=1}^T \|\mathbf{x}_{t,i^\star} - \mathbf{x}_t\|^2. \tag{by (D.1)}$$

Plugging the surrogate's empirical gradient variation back to the base regret, we obtain

$$\text{BASE-REG} \leq \frac{16G^2}{\lambda_{i^\star}} \ln\left(1 + 48L\lambda_{i^\star}\bar{F}_T + 2\lambda_{i^\star}^3 \sum_{t=1}^T \|\mathbf{x}_{t,i^\star} - \mathbf{x}_t\|^2\right) + \mathcal{O}(1)$$

$$\leq \frac{16G^2}{\lambda_{i^\star}} \ln\left(C_9\left(1 + 48L\lambda_{i^\star}\bar{F}_T\right)\right) + \frac{16G^2}{C_9\lambda_{i^\star}}\left(2\lambda_{i^\star}^3 \sum_{t=1}^T \|\mathbf{x}_{t,i^\star} - \mathbf{x}_t\|^2\right)$$

$$\leq \frac{32G^2}{\lambda}\ln(1 + 48L\bar{F}_T) + \frac{32G^2}{C_9}\sum_{t=1}^T \|\mathbf{x}_{t,i^\star} - \mathbf{x}_t\|^2 + \mathcal{O}(\ln C_9),$$

where the second step requires $C_9 \geq 1$ by Lemma 5 and uses the property of the best base learner, i.e., $\lambda_{i^\star} \leq \lambda \leq 2\lambda_{i^\star}$. The last step is due to $\lambda_i \leq 1$.

**Regret Analysis.** For *convex* functions, by combining the meta and base regret, it holds that

$$\text{REG}_T \leq C_0\sqrt{1 + 16D^2 L\bar{F}_T} + 5D\sqrt{1 + 16L\bar{F}_T} + C_1 \leq \mathcal{O}(\sqrt{F_T}),$$

where the last step is due to Lemma 9 of Zhao et al. [2024], restated below for self-containedness.

**Lemma 3** (Lemma 9 of Zhao et al. [2024]). *For any $x, y, a, b > 0$ satisfying $x - y \leq \sqrt{ax} + b$, it holds that $x - y \leq \sqrt{ay + ab} + a + b$.*

For *exp-concave* functions, by combining the meta and base regret, it holds that

$$\text{REG}_T \leq \left(\frac{C_0}{2C_3} + \frac{4G^2}{C_8} - \frac{\alpha_{i^\star}}{4}\right)\sum_{t=1}^T \langle \mathbf{g}_t, \mathbf{x}_t - \mathbf{x}_{t,i^\star}\rangle^2 + \frac{32d}{\alpha}\ln\left(1 + \frac{6L}{d}\bar{F}_T\right) + \mathcal{O}(C_3 + \ln C_8)$$

$$\leq \frac{32d}{\alpha}\ln\left(1 + \frac{6L}{d}\bar{F}_T\right) + \mathcal{O}(1) \leq \mathcal{O}\left(\frac{d}{\alpha}\ln F_T\right), \tag{by Lemma 6}$$

where the second step chooses $C_3 = \frac{4C_0}{\alpha_{i^\star}}$ and $C_8 = \max\{1, \frac{32G^2}{\alpha_{i^\star}}\}$. Note that such a parameter configuration will only add an $\mathcal{O}(1/\alpha)$ factor to the final regret bound, which can be absorbed.

For *strongly convex* functions, by combining the meta and base regret, it holds that

$$\text{REG}_T \leq \left(\frac{C_0 D^2}{2C_4} + \frac{32G^2}{C_9} - \frac{\lambda_{i^\star}}{4}\right)\sum_{t=1}^T \|\mathbf{x}_t - \mathbf{x}_{t,i^\star}\|^2 + \frac{32G^2}{\lambda}\ln(1 + 48L\bar{F}_T) + \mathcal{O}(C_4 + \ln C_9)$$

$$\leq \frac{32G^2}{\lambda}\ln(1+48L\bar{F}_T) \leq \mathcal{O}\left(\frac{1}{\lambda}\ln F_T\right), \qquad \text{(by Lemma 6)}$$

where the second step is by choosing $C_4 = \frac{4C_0 D^2}{\lambda_{i^\star}}$ and $C_9 = \max\{1, \frac{256G^2}{\lambda_{i^\star}}\}$. Note that such a parameter configuration will only add an $\mathcal{O}(1/\lambda)$ factor to the final regret bound, which can be absorbed. Also note that the constants $C_3, C_4, C_8, C_9$ only exist in analysis and thus can be chosen arbitrarily, finishing the proof. $\square$

## D.2 Proof of Theorem 3

*Proof.* To begin with, we give a different analysis of the empirical gradient variation:

$$\mathbb{E}[\bar{V}_T] \leq 5\mathbb{E}\left[\sum_{t=2}^{T}\|\nabla f_t(\mathbf{x}_t) - \nabla F_t(\mathbf{x}_t)\|^2\right] + 5\sum_{t=2}^{T}\|\nabla F_t(\mathbf{x}_t) - \nabla F_t(\mathbf{x}^\star)\|^2$$

$$+5\mathbb{E}\left[\sum_{t=2}^{T}\|\nabla F_t(\mathbf{x}^\star) - \nabla F_{t-1}(\mathbf{x}^\star)\|^2\right] + 5\sum_{t=2}^{T}\|\nabla F_{t-1}(\mathbf{x}^\star) - \nabla F_{t-1}(\mathbf{x}_{t-1})\|^2$$

$$+5\mathbb{E}\left[\sum_{t=2}^{T}\|\nabla F_{t-1}(\mathbf{x}_{t-1}) - \nabla f_{t-1}(\mathbf{x}_{t-1})\|^2\right] \leq 10\sigma_{1:T}^2 + 5\Sigma_{1:T}^2 + 20L\sum_{t=1}^{T}\mathcal{D}_{F_t}(\mathbf{x}^\star, \mathbf{x}_t), \quad \text{(D.2)}$$

where the first step is due to Cauchy-Schwarz inequality and the last step is because of the definitions of $\sigma_{1:T}^2$ and $\Sigma_{1:T}^2$ (given in Section 4) and the analysis proposed in Section 3.2.

In the following, we first give regret decompositions for different curvature types, then we analyze the meta and base regret, and combine them for the final regret guarantees.

**Regret Decomposition.** Denoting by $\mathbf{x}^\star \in \arg\min_{\mathbf{x}\in\mathcal{X}}\sum_{t\in[T]}f_t(\mathbf{x})$, for *convex* functions, we decompose the regret as

$$\mathbb{E}[\text{REG}_T] = \mathbb{E}\left[\sum_{t=1}^{T}F_t(\mathbf{x}_t) - \sum_{t=1}^{T}F_t(\mathbf{x}^\star)\right] = \mathbb{E}\left[\sum_{t=1}^{T}\langle\nabla F_t(\mathbf{x}_t), \mathbf{x}_t - \mathbf{x}^\star\rangle\right] - \sum_{t=1}^{T}\mathcal{D}_{F_t}(\mathbf{x}^\star, \mathbf{x}_t)$$

$$= \mathbb{E}\left[\sum_{t=1}^{T}\langle\nabla f_t(\mathbf{x}_t), \mathbf{x}_t - \mathbf{x}^\star\rangle\right] - \sum_{t=1}^{T}\mathcal{D}_{F_t}(\mathbf{x}^\star, \mathbf{x}_t)$$

$$= \underbrace{\mathbb{E}\left[\sum_{t=1}^{T}\langle\mathbf{g}_t, \mathbf{x}_t - \mathbf{x}_{t,i^\star}\rangle\right]}_{\text{META-REG}} + \underbrace{\mathbb{E}\left[\sum_{t=1}^{T}h_{t,i^\star}^{\text{c}}(\mathbf{x}_{t,i^\star}) - h_{t,i^\star}^{\text{c}}(\mathbf{x}^\star)\right]}_{\text{BASE-REG}} - \sum_{t=1}^{T}\mathcal{D}_{F_t}(\mathbf{x}^\star, \mathbf{x}_t),$$

where the first and third step use $F_t(\mathbf{x}) = \mathbb{E}[f_t(\mathbf{x})]$, the second step uses the definition of Bregman divergence, and the fourth step is due to $h_{t,i}^{\text{c}}(\mathbf{x}) \triangleq \langle\mathbf{g}_t, \mathbf{x}\rangle$.

For *exp-concave* functions, following the similar decomposition as in the proof of Theorem 1 in Appendix C, we decompose the regret as

$$\mathbb{E}[\text{REG}_T] = \mathbb{E}\left[\sum_{t=1}^{T}\langle\nabla F_t(\mathbf{x}_t), \mathbf{x}_t - \mathbf{x}^\star\rangle\right] - \frac{1}{2}\sum_{t=1}^{T}\mathcal{D}_{F_t}(\mathbf{x}^\star, \mathbf{x}_t) - \frac{1}{2}\sum_{t=1}^{T}\mathcal{D}_{F_t}(\mathbf{x}^\star, \mathbf{x}_t)$$

$$= \mathbb{E}\left[\sum_{t=1}^{T}\langle\nabla f_t(\mathbf{x}_t), \mathbf{x}_t - \mathbf{x}^\star\rangle\right] - \frac{1}{2}\sum_{t=1}^{T}\mathcal{D}_{f_t}(\mathbf{x}^\star, \mathbf{x}_t) - \frac{1}{2}\sum_{t=1}^{T}\mathcal{D}_{F_t}(\mathbf{x}^\star, \mathbf{x}_t)$$

$$\leq \mathbb{E}\left[\sum_{t=1}^{T}\langle\mathbf{g}_t, \mathbf{x}_t - \mathbf{x}^\star\rangle\right] - \frac{\alpha}{4}\sum_{t=1}^{T}\langle\mathbf{g}_t, \mathbf{x}_t - \mathbf{x}^\star\rangle^2 - \frac{1}{2}\sum_{t=1}^{T}\mathcal{D}_{F_t}(\mathbf{x}^\star, \mathbf{x}_t)$$

$$\leq \underbrace{\sum_{t=1}^{T}\langle\mathbf{g}_t, \mathbf{x}_t - \mathbf{x}_{t,i^\star}\rangle - \frac{\alpha_{i^\star}}{4}\sum_{t=1}^{T}\langle\mathbf{g}_t, \mathbf{x}_t - \mathbf{x}_{t,i^\star}\rangle^2}_{\text{META-REG}}$$

$$+ \mathbb{E}\left[\sum_{t=1}^{T} h_{t,i^\star}^{\exp}(\mathbf{x}_{t,i^\star}) - h_{t,i^\star}^{\exp}(\mathbf{x}^\star)\right] - \frac{1}{2}\sum_{t=1}^{T} \mathcal{D}_{F_t}(\mathbf{x}^\star, \mathbf{x}_t),$$

$$\underbrace{\phantom{+ \mathbb{E}\left[\sum_{t=1}^{T} h_{t,i^\star}^{\exp}(\mathbf{x}_{t,i^\star}) - h_{t,i^\star}^{\exp}(\mathbf{x}^\star)\right]}}_{\text{BASE-REG}}$$

where the second step uses the definition of the expected function $F_t(\cdot)$, the third step requires the exp-concavity of $f_t(\cdot)$, and the fourth step is due to $h_{t,i}^{\exp}(\mathbf{x}) \triangleq \langle \mathbf{g}_t, \mathbf{x} \rangle + \frac{\alpha_i}{4} \langle \nabla f_t(\mathbf{x}_t), \mathbf{x} - \mathbf{x}_t \rangle^2$, where $\alpha_i \in \mathcal{H}$, defined in (2.1).

For *strongly convex* functions, following the similar decomposition as in Appendix C, we have

$$\mathbb{E}[\text{REG}_T] = \mathbb{E}\left[\sum_{t=1}^{T} \langle \nabla F_t(\mathbf{x}_t), \mathbf{x}_t - \mathbf{x}^\star \rangle\right] - \frac{1}{2}\sum_{t=1}^{T} \mathcal{D}_{F_t}(\mathbf{x}^\star, \mathbf{x}_t) - \frac{1}{2}\sum_{t=1}^{T} \mathcal{D}_{F_t}(\mathbf{x}^\star, \mathbf{x}_t)$$

$$\leq \mathbb{E}\left[\sum_{t=1}^{T} \langle \mathbf{g}_t, \mathbf{x}_t - \mathbf{x}^\star \rangle\right] - \frac{\lambda}{4}\sum_{t=1}^{T} \|\mathbf{x}_t - \mathbf{x}^\star\|^2 - \frac{1}{2}\sum_{t=1}^{T} \mathcal{D}_{F_t}(\mathbf{x}^\star, \mathbf{x}_t)$$

$$\leq \underbrace{\sum_{t=1}^{T} \langle \mathbf{g}_t, \mathbf{x}_t - \mathbf{x}_{t,i^\star} \rangle - \frac{\lambda_{i^\star}}{4}\sum_{t=1}^{T} \|\mathbf{x}_t - \mathbf{x}_{t,i^\star}\|^2}_{\text{META-REG}}$$

$$+ \underbrace{\mathbb{E}\left[\sum_{t=1}^{T} h_{t,i^\star}^{\text{sc}}(\mathbf{x}_{t,i^\star}) - h_{t,i^\star}^{\text{sc}}(\mathbf{x}^\star)\right]}_{\text{BASE-REG}} - \frac{1}{2}\sum_{t=1}^{T} \mathcal{D}_{F_t}(\mathbf{x}^\star, \mathbf{x}_t),$$

where the second step, different from the exp-concave case, only requires the strong convexity of $F_t(\cdot)$, and the third step is due to $h_{t,i}^{\text{sc}}(\mathbf{x}) \triangleq \langle \mathbf{g}_t, \mathbf{x} \rangle + \frac{\lambda_i}{4}\|\mathbf{x} - \mathbf{x}_t\|^2$, where $\lambda_i \in \mathcal{H}$, defined in (2.1).

**Meta Regret Analysis.** Our Algorithm 1 can be applied to the SEA model without any algorithm modifications. As a result, we directly use the same parameter configurations as in the proof of Theorem 1 (i.e., in Appendix C).

For *convex* functions, the meta regret can be bounded as

$$\text{META-REG} \leq \mathbb{E}\left[C_0\sqrt{1 + D^2 \overline{V}_T} + C_1\right] \leq C_0\sqrt{1 + D^2 \mathbb{E}[\overline{V}_T]} + C_1$$

$$\leq C_0\sqrt{1 + 5D^2(2\sigma_{1:T}^2 + \Sigma_{1:T}^2) + 20D^2 L \sum_{t=1}^{T} \mathcal{D}_{F_t}(\mathbf{x}^\star, \mathbf{x}_t)} + C_1 \qquad \text{(by (D.2))}$$

$$\leq \mathcal{O}\left(\sqrt{\sigma_{1:T}^2 + \Sigma_{1:T}^2}\right) + \mathcal{O}(C_{10}) + \frac{C_0}{2C_{10}}\sum_{t=1}^{T} \mathcal{D}_{F_t}(\mathbf{x}^\star, \mathbf{x}_t),$$

where the second step is by Jensen's inequality and the last step is due to AM-GM inequality. $C_{10}$ is a constant to be specified.

For *exp-concave* and *strongly convex* functions, the meta regret is bounded in the same way as (C.5) and (C.6), and thus omitted here.

**Base Regret Analysis.** For *convex* functions, the base regret can be bounded as

$$\text{BASE-REG} \leq 5D\sqrt{1 + \mathbb{E}[\overline{V}_T]} \leq 5D\sqrt{1 + 10\sigma_{1:T}^2 + 5\Sigma_{1:T}^2 + 20L\sum_{t=1}^{T} \mathcal{D}_{F_t}(\mathbf{x}^\star, \mathbf{x}_t)}$$

$$\leq \mathcal{O}\left(\sqrt{\sigma_{1:T}^2 + \Sigma_{1:T}^2}\right) + \mathcal{O}(C_{11}) + \frac{5D}{2C_{11}}\sum_{t=1}^{T} \mathcal{D}_{F_t}(\mathbf{x}^\star, \mathbf{x}_t),$$

where the first step is by Jensen's inequality, the second step is due to (D.2), and the last step is because of AM-GM inequality. $C_{11}$ is a constant to be specified.

For *exp-concave* functions, the base regret is bounded by (C.7). Following (D.2), we control the empirical gradient variation defined on surrogates as

$$\mathbb{E}\left[\sum_{t=2}^{T}\left\|\nabla h_{t,i^\star}^{\exp}(\mathbf{x}_{t,i^\star}) - \nabla h_{t-1,i^\star}^{\exp}(\mathbf{x}_{t-1,i^\star})\right\|^2\right] \leq 3\mathbb{E}[\bar{V}_T] + 6\sum_{t=1}^{T}\left\|\frac{\alpha_{i^\star}}{2}\mathbf{g}_t\langle\mathbf{g}_t, \mathbf{x}_t - \mathbf{x}_{t,i^\star}\rangle\right\|^2$$

$$\leq 15(2\sigma_{1:T}^2 + \Sigma_{1:T}^2) + 60L\sum_{t=1}^{T}\mathcal{D}_{F_t}(\mathbf{x}^\star, \mathbf{x}_t) + 2\alpha_{i^\star}^2 G^2\sum_{t=1}^{T}\langle\mathbf{g}_t, \mathbf{x}_t - \mathbf{x}_{t,i^\star}\rangle^2.$$

Plugging the surrogate's empirical gradient variation back to the base regret, we obtain

$$\text{BASE-REG} \leq \frac{16d}{\alpha_{i^\star}}\ln\left(1 + \frac{15\alpha_{i^\star}}{8d}(2\sigma_{1:T}^2 + \Sigma_{1:T}^2) + \frac{15L\alpha_{i^\star}}{2d}\sum_{t=1}^{T}\mathcal{D}_{F_t}(\mathbf{x}^\star, \mathbf{x}_t)\right.$$

$$\left.+ \frac{\alpha_{i^\star}^3 G^2}{4d}\sum_{t=1}^{T}\langle\mathbf{g}_t, \mathbf{x}_t - \mathbf{x}_{t,i^\star}\rangle^2\right) \leq \mathcal{O}\left(\frac{d}{\alpha}\ln\left(\sigma_{1:T}^2 + \Sigma_{1:T}^2\right)\right) + \mathcal{O}(\ln C_{12})$$

$$+ \frac{120L}{C_{12}}\sum_{t=1}^{T}\mathcal{D}_{F_t}(\mathbf{x}^\star, \mathbf{x}_t) + \frac{4G^2}{C_{12}}\sum_{t=1}^{T}\langle\mathbf{g}_t, \mathbf{x}_t - \mathbf{x}_{t,i^\star}\rangle^2,$$

where the second step requires $C_{12} \geq 1$ by Lemma 5.

For *strongly convex* functions, we need to delve into the proof details of the base algorithm, i.e., OOMD (B.2) for strongly convex functions with step size $\eta_t = 2/(1 + \lambda_i t)$. For example, from Lemma 12 of Yan et al. [2023], the base regret can be bounded as

$$\text{BASE-REG} \leq 4\sum_{t=2}^{T}\frac{1}{\lambda_{i^\star}t}\mathbb{E}\left[\left\|\nabla h_{t,i^\star}^{\text{sc}}(\mathbf{x}_{t,i^\star}) - \nabla h_{t-1,i^\star}^{\text{sc}}(\mathbf{x}_{t-1,i^\star})\right\|^2\right] + \mathcal{O}(1).$$

Subsequently, we analyze the empirical gradient variation defined on surrogates in each round, i.e., $\|\nabla h_{t,i^\star}^{\text{sc}}(\mathbf{x}_{t,i^\star}) - \nabla h_{t-1,i^\star}^{\text{sc}}(\mathbf{x}_{t-1,i^\star})\|^2$. Denoting by $\sigma_t^2 \triangleq \max_{\mathbf{x}\in\mathcal{X}}\mathbb{E}_{f_t\sim\mathfrak{D}_t}[\|\nabla f_t(\mathbf{x}) - \nabla F_t(\mathbf{x})\|^2]$ and $\Sigma_t^2 \triangleq \mathbb{E}[\sup_{\mathbf{x}\in\mathcal{X}}\|\nabla F_t(\mathbf{x}) - \nabla F_{t-1}(\mathbf{x})\|^2]$ for simplicity,

$$\mathbb{E}\left[\left\|\nabla h_{t,i^\star}^{\text{sc}}(\mathbf{x}_{t,i^\star}) - \nabla h_{t-1,i^\star}^{\text{sc}}(\mathbf{x}_{t-1,i^\star})\right\|^2\right]$$

$$= \mathbb{E}\left[\left\|\mathbf{g}_t + \frac{\lambda_{i^\star}}{2}(\mathbf{x}_{t,i^\star} - \mathbf{x}_t) - \mathbf{g}_{t-1} - \frac{\lambda_{i^\star}}{2}(\mathbf{x}_{t-1,i^\star} - \mathbf{x}_{t-1})\right\|^2\right]$$

$$\leq 3\mathbb{E}\left[\|\mathbf{g}_t - \mathbf{g}_{t-1}\|^2\right] + 3\left\|\frac{\lambda_{i^\star}}{2}(\mathbf{x}_{t,i^\star} - \mathbf{x}_t)\right\|^2 + 3\left\|\frac{\lambda_{i^\star}}{2}(\mathbf{x}_{t-1,i^\star} - \mathbf{x}_{t-1})\right\|^2$$

$$\leq 15(\sigma_t^2 + \sigma_{t-1}^2 + 2L\mathcal{D}_{F_t}(\mathbf{x}^\star, \mathbf{x}_t) + 2L\mathcal{D}_{F_{t-1}}(\mathbf{x}^\star, \mathbf{x}_{t-1}) + \Sigma_t^2) \qquad \text{(by (D.2))}$$

$$+ \lambda_{i^\star}^2\|\mathbf{x}_{t,i^\star} - \mathbf{x}_t\|^2 + \lambda_{i^\star}^2\|\mathbf{x}_{t-1,i^\star} - \mathbf{x}_{t-1}\|^2,$$

where the first step is due to the property of the surrogate: $\nabla h_{t,i}^{\text{sc}}(\mathbf{x}_{t,i}) = \mathbf{g}_t + \frac{\lambda_i}{2}(\mathbf{x}_{t,i} - \mathbf{x}_t)$, and the second step is due to the Cauchy-Schwarz inequality. Plugging the above term back into the base regret and omitting the ignorable $\mathcal{O}(1)$ term, we achieve

$$\text{BASE-REG} \leq \frac{60}{\lambda_{i^\star}}\sum_{t=2}^{T}\frac{\sigma_t^2 + \sigma_{t-1}^2 + \Sigma_t^2}{t} + 120L\sum_{t=2}^{T}\frac{\mathcal{D}_{F_t}(\mathbf{x}^\star, \mathbf{x}_t) + \mathcal{D}_{F_{t-1}}(\mathbf{x}^\star, \mathbf{x}_{t-1})}{\lambda_{i^\star}t}$$

$$+ 4\sum_{t=2}^{T}\frac{\lambda_{i^\star}^2\|\mathbf{x}_{t,i^\star} - \mathbf{x}_t\|^2 + \lambda_{i^\star}^2\|\mathbf{x}_{t-1,i^\star} - \mathbf{x}_{t-1}\|^2}{\lambda_{i^\star}t},$$

To handle the above term of $\sum_{t=1}^{T}a_t/t$ for some variable sequence $\{a_t\}_{t=1}^{T}$, we import a useful lemma from Yan et al. [2023].

**Lemma 4** (Lemma 9 of Yan et al. [2023]). *For a sequence of $\{a_t\}_{t=1}^{T}$ and $b$, where $a_t, b > 0$ for any $t \in [T]$, denoting by $a_{\max} \triangleq \max_t a_t$ and $A \triangleq \lceil b\sum_{t=1}^{T}a_t\rceil$, we have*

$$\sum_{t=1}^{T}\frac{a_t}{bt} \leq \frac{a_{\max}}{b}(1 + \ln A) + \frac{1}{b^2}.$$

Using Lemma 4, we control the base regret as

$$
\text{BASE-REG} \leq \mathcal{O}\left(\frac{1}{\lambda}\left(\sigma_{\max}^2 + \Sigma_{\max}^2\right) \ln \frac{\sigma_{1:T}^2 + \Sigma_{1:T}^2}{\sigma_{\max}^2 + \Sigma_{\max}^2}\right)
$$

$$
+ \frac{480 LGD}{\lambda_{i^\star}} \ln\left(1 + 2\lambda_{i^\star} \sum_{t=1}^{T} \mathcal{D}_{F_t}(\mathbf{x}^\star, \mathbf{x}_t)\right) + \frac{8D^2}{\lambda_{i^\star}} \ln\left(1 + 2\lambda_{i^\star}^3 \sum_{t=1}^{T} \|\mathbf{x}_{t,i^\star} - \mathbf{x}_t\|^2\right)
$$

$$
\leq \mathcal{O}\left(\frac{1}{\lambda}\left(\sigma_{\max}^2 + \Sigma_{\max}^2\right) \ln \frac{\sigma_{1:T}^2 + \Sigma_{1:T}^2}{\sigma_{\max}^2 + \Sigma_{\max}^2}\right) + \mathcal{O}(\ln C_{13} + \ln C_{14})
$$

$$
+ \frac{960 LGD}{C_{13}} \sum_{t=1}^{T} \mathcal{D}_{F_t}(\mathbf{x}^\star, \mathbf{x}_t) + \frac{16D^2}{C_{14}} \sum_{t=2}^{T} \|\mathbf{x}_{t,i^\star} - \mathbf{x}_t\|^2,
$$

where the first term initializes Lemma 4 as $a_t = \sigma_t^2 + \sigma_{t-1}^2 + \Sigma_t^2$ (i.e., $a_{\max} = \mathcal{O}(\sigma_{\max}^2 + \Sigma_{\max}^2)$) and $b = 1/(\sigma_{\max}^2 + \Sigma_{\max}^2)$, the second term initializes Lemma 4 as $a_t = \mathcal{D}_{F_t}(\mathbf{x}^\star, \mathbf{x}_t) + \mathcal{D}_{F_{t-1}}(\mathbf{x}^\star, \mathbf{x}_{t-1})$ (i.e., $a_{\max} = 4GD$ due to Assumption 1) and $b = \lambda_{i^\star}$, the third term initializes Lemma 4 as $a_t = \lambda_{i^\star}^2 \|\mathbf{x}_{t,i^\star} - \mathbf{x}_t\|^2 + \lambda_{i^\star}^2 \|\mathbf{x}_{t-1,i^\star} - \mathbf{x}_{t-1}\|^2$ (i.e., $a_{\max} = 2D^2$ due to $\lambda_i \leq 1$ and Assumption 1) and $b = \lambda_{i^\star}$. The $\mathcal{O}(1)$ term contains ignorable terms like $\mathcal{O}(1/\lambda)$. The second step requires $C_{13}, C_{14} \geq 1$ by Lemma 5.

**Regret Analysis.** For *convex* functions, by combining the meta and base regret, it holds that

$$
\text{REG}_T \leq \mathcal{O}\left(\sqrt{\sigma_{1:T}^2 + \Sigma_{1:T}^2}\right) + \mathcal{O}(C_{10} + C_{11}) + \left(\frac{C_0}{2C_{10}} + \frac{5D}{2C_{11}} - 1\right) \sum_{t=1}^{T} \mathcal{D}_{F_t}(\mathbf{x}^\star, \mathbf{x}_t)
$$

$$
\leq \mathcal{O}\left(\sqrt{\sigma_{1:T}^2 + \Sigma_{1:T}^2}\right),
$$

by choosing $C_{10} = C_0$ and $C_{11} = 5D$.

For *exp-concave* functions, by combining the meta and base regret, it holds that

$$
\text{REG}_T \leq \mathcal{O}\left(\frac{d}{\alpha} \ln\left(\sigma_{1:T}^2 + \Sigma_{1:T}^2\right)\right) + \mathcal{O}(C_3 + \ln C_{12}) + \left(\frac{120L}{C_{12}} - \frac{1}{2}\right) \sum_{t=1}^{T} \mathcal{D}_{F_t}(\mathbf{x}^\star, \mathbf{x}_t)
$$

$$
+ \left(\frac{C_0}{2C_3} + \frac{4G^2}{C_{12}} - \frac{\alpha_{i^\star}}{4}\right) \sum_{t=1}^{T} \langle \mathbf{g}_t, \mathbf{x}_t - \mathbf{x}_{t,i^\star}\rangle^2 \leq \mathcal{O}\left(\frac{d}{\alpha} \ln\left(\sigma_{1:T}^2 + \Sigma_{1:T}^2\right)\right),
$$

by choosing $C_{12} = \max\{1, 240L, \frac{32G^2}{\alpha_{i^\star}}\}$ and $C_3 = \frac{4C_0}{\alpha_{i^\star}}$. Note that such a parameter configuration will only add an $\mathcal{O}(1/\alpha)$ factor to the final regret bound, which can be absorbed.

For *strongly convex* functions, by combining the meta and base regret, it holds that

$$
\text{REG}_T \leq \mathcal{O}\left(\frac{1}{\lambda}\left(\sigma_{\max}^2 + \Sigma_{\max}^2\right) \ln \frac{\sigma_{1:T}^2 + \Sigma_{1:T}^2}{\sigma_{\max}^2 + \Sigma_{\max}^2}\right) + \mathcal{O}(C_4 + \ln C_{13} + \ln C_{14})
$$

$$
+ \left(\frac{C_0 D^2}{2C_4} + \frac{16D^2}{C_{14}} - \frac{\lambda_{i^\star}}{4}\right) \sum_{t=1}^{T} \|\mathbf{x}_t - \mathbf{x}_{t,i^\star}\|^2 + \left(\frac{960 LGD}{C_{13}} - \frac{1}{2}\right) \sum_{t=1}^{T} \mathcal{D}_{F_t}(\mathbf{x}^\star, \mathbf{x}_t)
$$

$$
\leq \mathcal{O}\left(\frac{1}{\lambda}\left(\sigma_{\max}^2 + \Sigma_{\max}^2\right) \ln \frac{\sigma_{1:T}^2 + \Sigma_{1:T}^2}{\sigma_{\max}^2 + \Sigma_{\max}^2}\right),
$$

by choosing $C_{13} = \max\{1, 1920 LGD\}$, $C_{14} = \max\{1, \frac{128D^2}{\lambda_{i^\star}}\}$ and $C_4 = \frac{4C_0 D^2}{\lambda_{i^\star}}$. Note that such a parameter configuration will only add an $\mathcal{O}(1/\lambda)$ factor to the final bound, which can be absorbed.

Note that the constants $C_3, C_4, C_{10}, C_{11}, C_{12}, C_{13}, C_{14}$ only exist in analysis and thus can be chosen arbitrarily, finishing the proof. $\qquad\square$

# E   Technical Lemmas

**Lemma 5.** *For any $a > 1, b > 0$, it holds that $\ln(a + b) \leq \ln(Ca) + \frac{b}{C}$ for some $C \geq 1$.*

*Proof.* The one-line proof is presented below:

$$\ln(a + b) \leq \ln(Ca + b) \leq \ln(Ca) + \ln\left(1 + \frac{b}{Ca}\right) \leq \ln(Ca) + \frac{b}{C},$$

where the first step is due to $C \geq 1$, and the last step adopts $\ln(1 + x) \leq x$ for any $x \geq 0$. $\qquad \square$

**Lemma 6** (Corollary 5 of Orabona et al. [2012]). *If $a, b, c, d, x > 0$ satisfy $x - d \leq a \ln(bx + c)$, then it holds that*

$$x - d \leq a \ln\left(2ab \ln \frac{2ab}{e} + 2bd + 2c\right).$$

