# OpenReview forum: "A Simple and Optimal Approach for Universal Online Learning with Gradient Variations"
_NeurIPS.cc/2024/Conference — NeurIPS 2024 poster_

### Official Review · Reviewer_y66n · 2024-06-15

**Soundness:** 3
**Presentation:** 3
**Contribution:** 4
**Rating:** 7
**Confidence:** 4

**Summary:**

This paper studies universal Online Convex Optimization (OCO) with gradient-variation-dependent regret bounds. That is, to design one single algorithm that is unaware of but is able to adapt to both two following groundtruth: 1) the type of curvatures: the loss functions could be convex, strongly convex, or exp-concave; 2) the curvature coefficient: exp-concavity $\alpha$ or strong convexity $\lambda$. As a result, the regret guarantee achieved by the algorithm scales with the cumulative gradient variation $V_T$ (which depends on the loss function sequence), rather than the time horizon $T$, as well as the corresponding curvature type of the underlying loss functions.

This paper proposes a new simple algorithm, that for the first time achieves optimal gradient-variation bounds for all three curvature types. Note that the gradient-variation bounds immediately imply small-loss (aka first-order) regret bounds as well as worst-case bounds. The #base learners is also improved from $(\log T)^2$ to $\log T$ due to the two-layer structure. The main result also finds broad applications including the SEA model and dynamic regret bounds.

Technique-side, the improvement comes from an alternative way to analyze the the empirical gradient variation w.r.t. surrogate losses and utilizing a negative Bregmen divergence term (due to linearization) to cancel other positive terms, which is often omitted in the analysis.

**Strengths:**

I overall like such results. The authors present their observations and insights (from the regret analysis) in detail, leading to improved (and indeed optimal) regret bounds and even (conceptually) simpler algorithm design.

**Weaknesses:**

I didn’t spot any significant technical issues, and I'm just suggesting some minor “weakness”.

1. When the authors introduce the notion of $F_T$ and small-loss bound for the first time (around Eq. (1.2)), they may want to add that now the loss functions are non-negative (which I think should be necessary for all small-loss/first-order bounds?). Obviously, one can’t take squared root or logarithmic to a negative number.

2. In the application to dynamic regret, the problem setup is not clearly defined. What is the type of loss function? Is the strong-convexity/log-concavity known? It is particularly confusing since it’s right after the universal OCO setup.

**Questions:**

1. The idea of utilizing negative Bregmen divergence terms also appeared in other problems, such as high-probability regrets in adversarial bandits [1]. Could the authors comment on the connection (if any) between the use therein and this work?

2. Under the universal OCO setup, is it possible to handle time-varying curvatures, just like in [2]?

3. Seems that a universal OCO algorithm cannot be anytime? The reason is that the number of base learner (for discretization) depends on $T$.

References

[1] Lee, Chung-Wei, Haipeng Luo, Chen-Yu Wei, and Mengxiao Zhang. "Bias no more: high-probability data-dependent regret bounds for adversarial bandits and mdps." Advances in neural information processing systems 33 (2020): 15522-15533. https://arxiv.org/abs/2006.08040

[2] Luo, Haipeng, Mengxiao Zhang, and Peng Zhao. "Adaptive bandit convex optimization with heterogeneous curvature." In Conference on Learning Theory, pp. 1576-1612. PMLR, 2022. https://proceedings.mlr.press/v178/luo22a/luo22a.pdf

**Limitations:**

yes

---

> ### Author Rebuttal · Authors · 2024-08-06
>
> Thank you for the positive feedback and appreciation of our work! We answer your questions in the following.
>
> **Q1.** When the authors introduce the notion of $F_T$ and small-loss bound for the first time (around Eq. (1.2)), they may want to add that now the loss functions are non-negative (which I think should be necessary for all small-loss/first-order bounds?). Obviously, one can’t take squared root or logarithmic to a negative number.
>
> **A1.** Thank you for the advice. Actually, the non-negative assumption is not necessary. Specifically, the non-negativity is only used in the self-bounding property, i.e., $\|\nabla f(x)\|^2 \le 2L f(x)$ to achieve small-loss bounds. Without non-negativity, the self-bounding property still holds as $\|\nabla f(x)\|^2 \le 2L (f(x) - \inf_x f(x))$ [A Modern Introduction to Online Learning, Theorem 4.23]. In this case, we can correspondingly define a slightly complicated notion of small loss. We will emphasize that the non-negativity assumption is only adopted here for simplicity in the revised version.
>
> ---
>
> **Q2.** In the application to dynamic regret, the problem setup is not clearly defined. What is the type of loss function? Is the strong-convexity/log-concavity known? It is particularly confusing since it’s right after the universal OCO setup.
>
> **A2.** Thank you for the advice. Our results for dynamic regret minimization hold for ***convex*** functions, and we aim to validate the generality of our proposed technique in this application. We will clarify the problem setup more clearly in the revised version.
>
> ---
>
> **Q3.**  The idea of utilizing negative Bregmen divergence terms also appeared in other problems, such as high-probability regrets in adversarial bandits [1]. Could the authors comment on the connection (if any) between the use therein and this work?
>
> **A3.** Thank you for the reference. Lemma B.13 in [1] lower-bounds a quantity of $D_{\Psi}(u,x_t)$, where $u$ is the comparator and $x_t$ is the learner's decision. Their similarity to ours is that both work uses a negative Bregman divergence term for cancellation purposes. However, the difference is that their Bregman divergence is defined on a normal barrier $\Psi(\cdot)$, a special self-concordant barrier, which is a widely used concept in bandit convex optimization problems. In contrast, our Bregman divergence is defined on the online function $f_t(\cdot)$ and focuses on the full information setup.
>
> ---
>
> **Q4.** Under the universal OCO setup, is it possible to handle time-varying curvatures, just like in [2]?
>
> **A4.** Thank you for the insightful question. Our work and [2] investigate two orthogonal problems regarding function curvatures. Concretely, we study the case where the curvatures are ***unknown and homogeneous***, while [2] focuses on ***known but heterogeneous*** curvatures ('known' means that the learner can access the curvature coefficients after submitting her decisions). Handling time-varying curvatures in universal problems (i.e., ***both unknown and heterogeneous*** curvature) is extremely hard and it seems impossible to deal with such a challenging problem at the moment.
>
> ---
>
> **Q5.** Seems that a universal OCO algorithm cannot be anytime? The reason is that the number of base learner (for discretization) depends on $T$.
>
> **A5.** Thank you for the question. As the reviewer said, achieving anytime algorithms is hard since the number of base learners relies on the time horizon $T$. To handle unknown $T$ (an easier case), a possible solution that we could imagine may be using the doubling trick. However, it may import additional logarithmic factors in the final regret bound, which are unacceptable for exp-concave and strongly convex functions since $\log T$ factors would ruin the desired $\log V_T$ regret. This is also one of the open problems raised in the previous work of Zhang et al. [2022].

---

> ### Comment · Reviewer_y66n · 2024-08-08
>
> I thank the authors for the reply. Currently, I do not have any other concerns and I maintain my scores.

---

### Official Review · Reviewer_THmn · 2024-07-11

**Soundness:** 3
**Presentation:** 2
**Contribution:** 2
**Rating:** 5
**Confidence:** 3

**Summary:**

The paper studied the problem of regret minimization of a set of functions $\{f_t\}_{t=1}^{T}$ over a compact and convex constraint set $\mathcal{X}$, i.e.,
$\sum{t=1}^{T}f_{t}(x_t) - \text{min}{x\in\mathcal{X}}\sum{t=1}^{T}f_{t}(x),$
where $x_t$ is the output of the proposed algorithm at round $1\leq t\leq T$.
The set of functions ${f_t}_{t=1}^{T}$ potentially satisfy certain curvature assumptions, e.g., strong convexity, convexity, or exp-concavity. In the paper, it is unknown which curvature assumption the function satisfies. The main goals of the paper are the following:

1. To construct a universal algorithm that adaptively acts on the curvature property of the function and achieves a proper regret bound.
2. For the case where the function is $\lambda$-strongly convex or $\alpha$-exp-concave, the algorithm should be adapted with respect to the curvature parameter, $\lambda$ to $\alpha$.
3. The algorithm should achieve a good problem-dependent regret bound: The goal of the paper is to attain a regret bound that depends on the following quantities:
$V_T = \sum_{t=1}^{T}\text{sup}_{x\in \mathcal{X}} \| \nabla f_{t}(x) - \nabla_f{t-1} (x) \|^2, \quad \text{and}\quad F_T = \text{min}{x\in\mathcal{X}}\sum{t=1}^{T}f_t(x).$

The proposed algorithm of the paper is a modification of the algorithm proposed by [1]. Similar to the approach introduced by [1], in Algorithm 1 (page 5) of the paper, the authors proposed $\emph{base learners}$ that are aggregated by a meta-algorithm, which outputs the final output of the algorithm at round $t$, $x_t$. The contribution of the paper mainly concerns the technical aspects that outperform the performance of [1] from the following points of view:

1.The paper improves the number of required base learners from $\log(T)^2$ (in [1]) to $\log(T)$.

2.This improvement of the algorithm outperforms the algorithm proposed by [1] up to a logarithmic factor for the situation where the loss functions $f_t$ are convex.

[1] Y.-H. Yan, P. Zhao, and Z.-H. Zhou. Universal online learning with gradual variations: A multi-layer online ensemble approach. In Advances in Neural Information Processing Systems 36 (NeurIPS), 2023.

**Strengths:**

The paper uses simple but interesting technique that contributes in tigher bounds for the case of convex losses. Inspired by [2] the authors used that exploit the imposed smoothness assumption of the loss function and Bregman divergence negative term from linearization of loss function, explained in Section 3.2.



[2] P. Joulani, A. Raj, A. Gyorgy, and C. Szepesvari. A simpler approach to accelerated optimization: iterative averaging meets optimism. In Proceedings of the 37th International Conference on Machine Learning (ICML), 2020.

**Weaknesses:**

The main weakness of the paper lies in its presentation. The content is too dense, and the last section on dynamic regret could be moved to the appendix. Some key parts of the paper are not well explained. For instance, it is unclear how the authors managed to outperform the number of required base learners in [1] by a logarithmic factor. Was this achieved through the application of a Bregman divergence negative term?

The contribution of the paper is limited to a simple technical improvement that enhances the achieved regret up to a logarithmic factor for convex functions.

The optimality of the result with respect to $V_T$ and $F_T$ has not been discussed by the authors.


[1] Y.-H. Yan, P. Zhao, and Z.-H. Zhou. Universal online learning with gradual variations: A multi-layer online ensemble approach. In Advances in Neural Information Processing Systems 36 (NeurIPS), 2023.

**Questions:**

1. I do not understand the comment on the small $\alpha$ and $\lambda$ in lines 146-147. Can these cases be considered convex? For the convex case, the rate is of the order of $\sqrt{T}$. How do the optimal minimax results hold for this regime, which indicates that the regret is linear?

2. I would appreciate it if the authors could explain the question I raised in the weakness section and outline the main difference that helps them improve the number of base learners.

3. I do not understand the comment in line 305: "which can be easily canceled by the negative term from curvatures in the meta regret." For this cancellation to occur, the coefficient in the equation after line 304 really matters. Could the authors explain this further?

4. Could the authors explain if the final result is optimal with respect to $V_T$ and $F_T$?

Minor Points:

The terms $\sigma_{\text{\max}}$ and $\Sigma_{\text{\max}}$ are not defined in Theorem 3.

**Limitations:**

The authors adequately addressed the limitations.

---

> ### Author Rebuttal · Authors · 2024-08-06
>
> Thank you for the positive feedback and appreciation of our work! Due to the 6,000-character limit of the rebuttal, we address your major questions below and respond to other minor issues in the next reply after the discussion period starts.
>
> ---
>
> **Q1.** I do not understand the comment on the small $\alpha$ and $\lambda$ in lines 146-147. Can these cases be considered convex? For the convex case, the rate is of the order of $\sqrt{T}$. How do the optimal minimax results hold for this regime, which indicates that the regret is linear?
>
> **A1.** Thank you for the question. We take $\alpha$-exp-concavity as an example for illustration. Since **exp-concave functions are also convex**, universal methods actually guarantee the final rate as $\min\\{\frac{d}{\alpha} \log V_T, \sqrt{V_T}\\}$, thus safeguarding $\sqrt{V_T}$ even when $\alpha = 1/T$. Similarly, a bound of $\min\\{\frac{d}{\alpha} \log T, \sqrt{T}\\}$ holds for minimax regret for exp-concave functions. We will discuss this issue in the next version. Thank you for highlighting it for improvement.
>
> ---
>
> **Q2.** It is unclear how the authors managed to outperform the number of required base learners in [1] by a logarithmic factor. Was this achieved through the application of a Bregman divergence negative term?
>
> **A2.** Thank you for the question. We clarify that the ***base learner number is only determined by the algorithm design***. Previous $\log^2 T$ base learners come from their *multi-layer (three-layer)* algo. with a *two-layer meta learner*, as explained in Line 90 and Lines 233-237. By contrast, we have only $\log T$ base learners by using ***the single Optimistic ADAPT-ML-PROD as the meta learner***, resulting in *two layers (not three)*. We clarify that simply using Optimistic ADAPT-ML-PROD in previous methods (e.g., in [3]) cannot achieve the optimal rates. The simplicity of our meta algo. is owing to our novel analysis to bypass the stability cancelation in [3].
>
> ---
>
> **Q3.** I do not understand the comment in line 305: "which can be easily canceled by the negative term from curvatures in the meta regret." For this cancellation to occur, the coefficient in the equation after line 304 really matters. Could the authors explain this further?
>
> **A3.** Thank you for the insightful observation. We have chosen appropriate coefficients in the detailed proofs in the appendices and we omitted them from the main paper only for clarity. For more details, please kindly refer to the proof of Theorem 1. For example, in the 'Regret Analysis' part on Page 17, the coefficients from $C_2$ to $C_7$ are used to carefully balance the positive and negative terms such that they can be canceled. And we clarify that these coefficients only exist in analysis and thus can be chosen arbitrarily.
>
> ---
>
> **Q4.** Could the authors explain if the final result is optimal with respect to $V_T$ and $F_T$?
>
> **A4.** Thank you for the great question. We take $V_T$-bounds as an example for clarification.
>
> * For known curvatures (strongly convex/exp-concave/convex), the SOTA rates are $\log V_T$, $d \log V_T$ and $\sqrt{V_T}$ , and can recover minimax optimal $\log T$, $d \log T$ and $\sqrt{T}$. For convex functions, an $\Omega(\sqrt{V_T})$ lower bound is known in [Regret Bounded by Gradual Variation for Online Convex Optimization, Remark 5]. Nevertheless, for the other two cases, though the current rates are believed to be optimal, the precise problem-dependent lower bounds are still unclear.
> * In our paper, for "optimality", we mean that our universal bounds match the same best-known rates as if the curvature information were known. We will revise the paper to make this point clearer to readers.
>
> ---
>
> **Q5.** The contribution of the paper is limited to a simple technical improvement that enhances the achieved regret up to a logarithmic factor for convex functions.
>
> **A5.** We would like to take this opportunity to further highlight our contributions, including the problem, techniques, and applications. We will improve the presentation in the revised version.
>
> * **Problem:** Due to the importance of universal online learning due to its robustness and gradient variation due to its profound connections with stochastic/adversarial optimization, game theory, etc. Studying how to achieve gradient-variation regret in universal online learning is essential. We contribute to ***achieving the optimal rates in this fundamental problem***, thereby solving the ***major open problem in [3]*** (please kindly refer to their conclusion).
> * **Techniques:** ***Our technique's simplicity is advantageous for its generality***. It succeeds in avoiding controlling the algorithm stability of $\|x_t - x_{t-1}\|^2$, the reason for the suboptimality and inefficiency of [3]. It is ***the first alternative solution to gradient-variation regret since it was first proposed in [1]***. We believe this technical insight is useful for broader applications and take ***dynamic regret minimization*** as an example, where a much simpler method using our technique obtains the same SOTA dynamic regret as [2].
> * **Applications:** We validate the significance of our results in the ***stochastically extended adversarial (SEA) model*** and ***dynamic regret minimization*** and achieve ***SOTA*** rates therein. Note that [3] only achieved suboptimal bounds in the SEA model (Table 3 in [3]) and cannot be used in dynamic regret minimization.
>
> ---
>
> We will carefully revise the paper according to your suggestions and questions. If our responses have satisfactorily addressed your concerns, we would appreciate it so much if you could re-evaluate our work. Thank you!
>
> **References:**
>
> [1] Online Optimization with Gradual Variations, COLT 2012 (Best Student Paper)
>
> [2] Adaptivity and Non-stationarity: Problem-dependent Dynamic Regret for Online Convex Optimization, JMLR 2024
>
> [3] Universal Online Learning with Gradient Variations: A Multi-layer Online Ensemble Approach, NeurIPS 2023

---

> > ### Author Response · Authors · 2024-08-08
> > **Response to other comments**
> >
> > In this reply, we respond to presentation/definition issues, hoping to help you better understand our work.
> >
> > **Q6.** The content is too dense, and the last section on dynamic regret could be moved to the appendix.
> >
> > **A6.** Thank you for the advice. We use dynamic regret minimization as an example to validate the generality and significance of our technique and believe it is worth listing in the main part. Given that an extra page is allowed in the camera-ready version, we will provide more detailed explanations of our method and applications to help readers understand them better.
> >
> > **Q7.** The terms $\sigma_{\max}$ and $\Sigma_{\max}$ are not defined in Theorem 3.
> >
> > **A7.** Thank you for pointing it out. $\sigma_{\max}^2 \triangleq \max_{t \in [T]} \max_{\mathbf{x} \in \mathcal{X}} \mathbb{E}\_{f_t \sim \mathcal{D}\_t} [\\|\nabla f_t(\mathbf{x}) - \nabla F_t(\mathbf{x})\\|^2]$ and $\Sigma_{\max}^2 \triangleq \max_{t \in [T]} \sup_{\mathbf{x} \in \mathcal{X}} \\|\nabla F_t(\mathbf{x}) - \nabla F_{t-1}(\mathbf{x})\\|^2$. We will add their definitions in the revised version.

---

> > > ### Comment · Reviewer_THmn · 2024-08-08
> > > **Rebuttal acknowledgment**
> > >
> > > I would like to acknowledge the author's rebuttal. For now, I will maintain my current score. I plan to discuss the paper with the other reviewers and look forward to the author's discussions with them as well. I will update my score accordingly.

---

> > > > ### Author Response · Authors · 2024-08-11
> > > > **Thanks for the feedback**
> > > >
> > > > Thanks for the feedback. We will work on improving the presentation and incorporating the above clarifications in the revised version to ensure that readers gain a better understanding of our contributions. Please let us know if you have any more questions, and we are happy to provide further clarification.

---

### Official Review · Reviewer_S4Zf · 2024-07-11

**Soundness:** 3
**Presentation:** 3
**Contribution:** 2
**Rating:** 5
**Confidence:** 3

**Summary:**

This paper investigates the problem of universal online convex optimization to achieve problem dependent regret guarantees for different classes of convex functions (strongly convex, exp-concave, and convex) simultaneously. Problem/function/data dependent regret guarantees have become popular in literature to bridge stochastic and adversarial guarantees.

**Strengths:**

S1) The paper is well written and easy to understand.

S2) The literature review is comprehensive and up to date.

S3) Simplicity of the incorporation of Bregman divergence is a plus.

**Weaknesses:**

W1) The contribution seems limited in that the improvement is only logarithmic for both efficiency and regret results.

W2) While the regret analysis is novel, algorithmic contribution is very limited, which leads me to believe this paper is more suitable to be a technical note.

**Questions:**

Q1) Why is $\log^2 T$ computational complexity claimed to be inefficient throughtout the paper? In Table 1, the number of gradient queries and base learners are given as part of efficiency, however, a decrease on the number of queries seems much more significant to me.

Q2) Improvement over the results of Yan et al. [2023] seems incremental. Are there scenarios where this improvement becomes significant?

Q3) Is your approach the same as Zhang et al. [2022]  but using Optimistic ADAPT-ML-PROD [Wei et al., 2016] instead of  ADAPT-ML-PROD [Gaillard et al., 2014]?

**Limitations:**

No significant limitations. Necessary assumptions about the problem setting are properly addressed.

---

> ### Author Rebuttal · Authors · 2024-08-06
>
> We sincerely appreciate your feedback. Below, we aim to address your concerns about the number of base learners, the significance of our contributions, and the algorithmic improvements.
>
> ---
>
> **Q1-a.** Why is $\log^2 T$ computational complexity claimed to be inefficient throughout the paper?
>
> **A1-a.** Thank you for the question. We clarify that the $\log^2 T$ complexity is ***less efficient*** compared with methods with only $\log T$ base learners. We will revise the statements of "inefficient" in the next version.
>
> **Q1-b.** In Table 1, the number of gradient queries and base learners are given as part of efficiency, however, a decrease on the number of queries seems much more significant to me.
>
> **A1-b.** Thank you for the comment. Online ensemble runs a meta learner over multiple base learners and each base learner runs an online learning algo. separately. Consider a simple case: each base learner runs OGD $x_{t+1} = \Pi_{D}[x_t - \eta \nabla f_t(x_t)]$, where $\nabla f_t(x_t)$ is the gradient and $\Pi_D[\cdot]$ is the projection onto the domain. Since computing gradients and projections is usually time-consuming, and each base learner has to conduct such operations in their own update, it is essential to reduce the number of them. In this work, we reduce the gradient query number to one per round by using surrogate functions and reduce the projection number by reducing the number of base learners. We will revise the paper to make this clearer to readers.
>
> ---
>
> **Q2.** The contribution seems limited in that the improvement is only logarithmic for both efficiency and regret results. Are there scenarios where the improvement over the results of Yan et al. [2023] becomes significant? (response to W1 and Q2)
>
> **A2.** Thank you for the question. We clarify the significance of our paper from the aspects of the problem, applications, and techniques below.
>
> * **Problem:** OCO is fundamental in online learning due to its generality. Curvatures are important to the best attainable results in OCO. However, traditional methods require knowing them in advance to select suitable algorithms, which is cumbersome in practice. ***Universal*** methods do not require such prior knowledge and can achieve the same optimal rates as if the curvatures were known. Furthermore, the ***gradient variation*** is essential in modern online learning due to its profound connections with stochastic/adversarial optimization, game theory, etc. Therefore, studying how to achieve gradient-variation regret in universal online learning is an essential problem. We contribute to ***achieving the optimal rates in this fundamental problem***, thereby solving the ***major open problem in [3]*** (please kindly refer to their conclusion).
> * **Applications:** We validate the significance of our results in the ***stochastically extended adversarial (SEA) model*** and ***dynamic regret minimization*** and achieve ***SOTA*** rates therein. Note that [3] only achieved suboptimal bounds in the SEA model (Table 3 therein) and cannot be used in the dynamic regret minimization problem.
> * **Techniques:** We have provided a novel technical perspective to gradient-variation regret. Instead of controlling the algorithm stability of $\|x_t - x_{t-1}\|^2$, which is the reason for the suboptimality and inefficiency of [3]'s method, we propose leveraging the Bregman divergence-related negative terms. This is ***the first alternative solution to gradient-variation regret since it was first proposed in [1]***. We believe this technical insight is useful for broader applications and take ***dynamic regret minimization*** as an example, where a much simpler method using our technique obtains the same SOTA dynamic regret as [2]. It cannot be done via the techniques of [3].
>
> At last, we clarify that our improvement in efficiency is not just reducing previous ones by a log factor, but ***simplifying the previous complicated three-layer algorithm of [3] to a simpler two-layer one***, as explained in Lines 223-237.
>
> ---
>
> **Q3.** While the regret analysis is novel, algorithmic contribution is very limited. Is your approach the same as Zhang et al. [2022] but using Optimistic ADAPT-ML-PROD [Wei et al., 2016] instead of ADAPT-ML-PROD [Gaillard et al., 2014]? (response to W2 and Q3)
>
> **A3.** Thank you for the question. Below, we clarify the differences between our method and that of Zhang et al. [2022].
>
> * **Meta Algorithm:** We use Optimistic ADAPT-ML-PROD while they used ADAPT-ML-PROD. We clarify that simply using Optimistic ADAPT-ML-PROD in previous methods (e.g., in [3]) will ***not*** lead to the same optimal rates as we did. The simplicity of our meta algorithm is owing to our novel analysis to bypass the stability cancelation in [3].
> * **Base Algorithm:** We use ***surrogate functions***, defined in eq. (3.1), for the base learner update in Algorithm 1 (they directly optimized $f_t(\cdot)$). This is also the key to ensuring our method requires only one gradient query per round.
>
> Besides, as shown in our title, one of our contributions is proposing a *simple and optimal* algorithm. In our humble opinion, designing a simple and optimal algorithm is also an unignorable contribution, because ***simple algorithms are usually more efficient and reflect the essence of the problem***, which is also acknowledged by Reviewer #y66n as one of our contributions.
>
> ---
>
> We will carefully revise the paper to ensure our contributions are clear to readers. If our responses have properly addressed your concerns, we would appreciate it so much if you could re-evaluate our work. Thank you!
>
> **References:**
>
> [1] Online Optimization with Gradual Variations, COLT 2012 (Best Student Paper)
>
> [2] Adaptivity and Non-stationarity: Problem-dependent Dynamic Regret for Online Convex Optimization, JMLR 2024
>
> [3] Universal Online Learning with Gradient Variations: A Multi-layer Online Ensemble Approach, NeurIPS 2023

---

> > ### Comment · Reviewer_S4Zf · 2024-08-09
> >
> > I acknowledge the author's rebuttal. Thank you for a detailed response. I suggest you revise the manuscript with these explanations. I have no further questions.

---

> > > ### Author Response · Authors · 2024-08-11
> > > **Thanks for the feedback**
> > >
> > > We appreciate your feedback and acknowledgment of our work. We will emphasize more about the computational efficiency, the significance of our improvements over Yan et al. [2023], and the algorithm contributions in the revised version. Thanks again for your valuable and helpful review.

---

### Official Review · Reviewer_KThz · 2024-07-17

**Soundness:** 3
**Presentation:** 3
**Contribution:** 3
**Rating:** 6
**Confidence:** 3

**Summary:**

The authors study the regret minimization problem in online convex optimization without access to curvature information. They tackle the task of achieving problem-dependent optimal regret while requiring no prior knowledge of the function class (convex, exp-concave, or strongly convex). They propose an efficient-to-implement two-layer online ensemble structure that requires only one gradient query within each round. Their main technical novelty lies in providing a novel approach for gradient-variation bounds.

**Strengths:**

The authors tackle a very interesting problem in online convex optimization. The paper is well-written and the presentation makes it easy to follow. The main novelty lies in Sections 3.2 and 3.3, where they provide a new way of tackling gradient variations by utilizing the Bregman divergence term. They also make clever use of Proposition 1 in their analysis. The overall method utilizes techniques from several existing works and cleverly combines them to achieve an impressive bound on the regret.

**Weaknesses:**

The proposed approach seems reasonable to me. While I have not gone through the technical details very carefully, I seek one clarification on the proof of Theorem 1. In my opinion, the bottleneck of the proof is in showing the existence of an appropriate choice of $C_3$ and $C_4$ (page 17, line 594, 596). Can the authors comment if such a setting always exists? I would at least expect certain conditions like $\alpha_i^* > G^2/9L $ or $\lambda_i^* > 1/9L$ for results to hold.

Another small thing: I understand the authors ignore very small terms like $\log \log T$ from the order notation. It might be good to put a note in the introduction about it while presenting the result. I understand that it is there in Section 3.1 -- it might be good to move it earlier.

**Questions:**

See above.

**Limitations:**

See above.

---

> ### Author Rebuttal · Authors · 2024-08-06
>
> Thank you for the positive feedback and appreciation of our work! In the following, we answer your questions about feasible analytical parameters and the statements of the $\log \log T$ term.
>
> ---
>
> **Q1.** In my opinion, the bottleneck of the proof is in showing the existence of an appropriate choice of $C_3$ and $C_4$ (page 17, line 594, 596). Can the authors comment if such a setting always exists? I would at least expect certain conditions like $\alpha_i^* > G^2 / 9L$ or $\lambda_i^* > 1/9L$ for results to hold.
>
> **A1.** Thank you for the great question! We checked the proof carefully again and realized that the current choices of these parameters may not be favorable enough. To fix this, we provide another choice below.
>
> For simplicity, we omit unimportant constants (we omit $C_0$ also since $C_0 \approx \log \log T$) and consider satisfying the following two conditions simultaneously: $\frac{1}{C_3} + \frac{1}{C_6} - \alpha_i^* \le 0$ and $\frac{1}{C_6} - 1 \le 0$. The new choice of parameters is $C_6 = \max\{1, \frac{2}{\alpha_i^*}\}$ and $C_3 = \frac{2}{\alpha_i^*}$ such that $\frac{1}{C_6} \le \frac{\alpha_i^*}{2}$, $\frac{1}{C_6} \le 1$, and $\frac{1}{C_3} = \frac{\alpha_i^*}{2}$. Note that these constants only exist in analysis and thus can be chosen arbitrarily. As a cost, we need to analyze the positive terms depending on $C_3$ and $C_6$, i.e., $O(C_3 + \ln C_6)$, as shown in the equation above Line 594. Fortunately, the positive terms are also acceptable since $C_3 = \frac{2}{\alpha_i^*} \approx \frac{1}{\alpha}$ and $\ln C_6 \approx \ln (1+\frac{1}{\alpha}) \le \frac{1}{\alpha}$. And $\frac{1}{\alpha}$ is a constant independent of time horizon $T$ and can be absorbed into the final bound of $\frac{d}{\alpha}\log V_T$. We will use this new choice of analytical parameters in the revised version. Thanks for highlighting this point for improvement.
>
> ---
>
> **Q2.** I understand the authors ignore very small terms like $\log \log T$ from the order notation. It might be good to put a note in the introduction about it while presenting the result. I understand that it is there in Section 3.1 -- it might be good to move it earlier.
>
> **A2.** Thank you for the advice. We will add a statement about the $\log \log T$ term in the caption of Table 1 in the revised version. Thanks for highlighting this point for improvement.
>
> ---
>
> We believe that our work offers valuable contributions to the community. We hope our response addresses your concerns and we are happy to provide further clarifications if needed during the following author-reviewer discussions.

---

> > ### Comment · Reviewer_KThz · 2024-08-10
> >
> > Thank you for your response. The new settings of parameters seem to work well. Could you comment on a couple of more things :
> > 1. The construction of the algorithm is dependent on knowing the time horizon $T$. Is it possible to extend this to scenarios when the time horizon is not known?
> > 2. The extension to dynamic regret bound seems interesting to me. If I am correct this part does not utilize any curvature information, likely due to $P_T$ and $V_T$ being the dominant terms in the final expression. Is that correct?  Also, in Theorem 5, is there an $\mathcal{O}(V_T)$ term missing from the final expression?

---

> > > ### Author Response · Authors · 2024-08-11
> > > **Response to the follow-up questions**
> > >
> > > Thanks for your feedback and the follow-up questions. Below, we answer your questions about the unknown time horizon and the dynamic regret bound, hoping to help you gain a better understanding of our work.
> > >
> > > ---
> > >
> > > **Q3.** The construction of the algorithm is dependent on knowing the time horizon $T$. Is it possible to extend this to scenarios when the time horizon is not known?
> > >
> > > **A3.** Thanks for this insightful question. As shown in Eq. (2.1) on Page 4, setting the number of base learners requires knowing $T$ in advance. To remove this dependence, a possible solution that we could imagine is the ***doubling trick***. However, this would possibly import an additional $\log T$ factor to the final regret bound, which would ruin the desired $\log V_T$ rate for exp-concave and strongly convex functions. Indeed, as explained in the response **A5** for Reviewer #y66n, making our algorithm anytime is a challenging open problem worth exploring in future work. This challenge is also noted in previous studies of universal online learning like Zhang et al. [2022].
> > >
> > > ---
> > >
> > > **Q4.** The extension to dynamic regret bound seems interesting to me. If I am correct, this part does not utilize any curvature information, likely due to $P_T$ and $V_T$ being the dominant terms in the final expression. Is that correct? Also, in Theorem 5, is there an $O(V_T)$ term missing from the final expression?
> > >
> > > **A4.** Thank you for the great question. Below, we answer your two questions separately.
> > >
> > > * **Curvature Information:** In this part, we aim to validate the effectiveness and generality of our technique in dynamic regret minimization for ***convex*** functions. It means that we do ***not*** consider curvature information such as exp-concavity or strong convexity in this part. We will make more clarifications about the problem setup of this part in the revised version.
> > > * **Missing $O(V_T)$:** In Theorem 5, there is no missing $O(V_T)$ term, and our obtained $O(\sqrt{(1+V_T+P_T)(1+P_T)})$ dynamic regret has matched the state-of-the-art rate, see Theorem 4 in [1].
> > >
> > > [1] Adaptivity and Non-stationarity: Problem-dependent Dynamic Regret for Online Convex Optimization, JMLR 2024
> > >
> > > ---
> > >
> > > We greatly appreciate your thorough review and helpful feedback. We will incorporate the above clarifications in the revised version. Please let us know if you have any more questions, and we are happy for further discussions.

---

### Decision · Program_Chairs · 2024-09-25

**Decision:**

Accept (poster)

**Comment:**

Overall all reviewers were positive about this work and I support their decision.